# Compositional Visual Planning via Inference-Time Diffusion Scaling

**Yixin Zhang**[1,*] **Yunhao Luo**[1,2], **Utkarsh A. Mishra**[1], **Woo Chul Shin**[1],
**Yongxin Chen**[1], **Danfei Xu**[1]
[1]Georgia Institute of Technology
[2]University of Michigan

## Abstract

Diffusion models excel at short-horizon robot planning, yet scaling them to long-horizon tasks remains challenging due to computational constraints and limited training data. Existing compositional approaches stitch together short segments by separately denoising each component and averaging overlapping regions. However, this suffers from instability as the factorization assumption breaks down in noisy data space, leading to inconsistent global plans. We propose that the key to stable compositional generation lies in enforcing boundary agreement on the estimated clean data (Tweedie estimates) rather than on noisy intermediate states. Our method formulates long-horizon planning as inference over a chain-structured factor graph of overlapping video chunks, where pretrained short-horizon video diffusion models provide local priors. At inference time, we enforce boundary agreement through a novel combination of synchronous and asynchronous message passing that operates on Tweedie estimates, producing globally consistent guidance without requiring additional training. Our training-free framework demonstrates significant improvements over existing baselines, effectively generalizing to unseen start-goal combinations that were not present in the original training data. Project website: https://comp-visual-planning.github.io/

## 1 Introduction

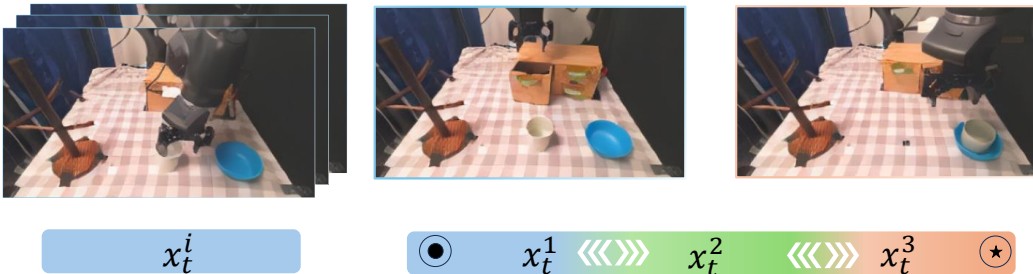

Training as a Unified Factor       Inference as a Chained-Structured Factor Graph

⦿ Start Boundary Variable    ⭐ Goal Boundary Variable    ⟪⟩⟫ Transition Boundary Variable

Figure 1: **Compositional Visual Planning via Inference Time Diffuser Scaling.** We train a short-horizon visual diffusion model on clips treated as a single factor. At inference, we scale visual planning horizon without retraining by chaining overlapping factors into a linear factor graph: the start and goal boundary variables are anchored at the ends, while neighboring factors exchange information through shared transition boundary variables.

Generative diffusion models have shown strong capacity for modeling complex, high-dimensional distributions over images, videos, and robot plans. In planning, they offer a compelling alternative to per-instance optimization: instead of solving a new search problem for every start–goal pair, we can sample likely solutions from a learned generator. However, extending video-based planning to long horizons remains challenging: most backbones are trained on short clips, compute and memory

---

*Correspondence to: yzhang4179@gatech.edu

scale unfavorably with sequence length, and long-range constraints (contacts, object persistence, and start–goal satisfaction) must be maintained throughout the rollout.

Classical planning methods, such as Task and Motion Planning, decompose tasks into structured subproblems and enforce constraint satisfaction through symbolic operators, while hierarchical control methods first solve for a high-level task plan and then refine it into a low-level motion plan. Compositional diffusion planning follows these core principles but provides a data-driven, probabilistic alternative to hand-engineered classical and hierarchical planners. We adopt a compositional generation perspective on long-horizon planning, we compose plans from overlapping, short-horizon factors produced by a pretrained diffusion model. The central challenge is the consistency of this composition: during forward diffusion, noisy variables become entangled across time, breaking factorization assumptions behind common compositional heuristics (e.g., score averaging) (Zhang et al., 2023; Mishra et al., 2023; 2024; Bar-Tal et al., 2023) and yielding brittle behavior when long-range constraints must propagate.

Our key insight is to compose where diffusion model estimations are most reliable: on their Tweedie estimates, which provide a stable domain in which strong and explicit compositional heuristics can be applied. We formulate planning as inference in a chain-structured factor graph over overlapping video chunks. Local priors come from a short-horizon diffusion backbone; global coherence is enforced by boundary agreement on Tweedie predictions, not on noisy states. We instantiate this approach for compositional visual diffusion planning, representing a plan as a sequence of images. Crucially, the method operates purely at inference time: the short-horizon diffusion backbone is trained once on short clips and then frozen; at test time we compose long-horizon plans via message passing on Tweedie estimates, with no additional training, fine-tuning, or task-specific adapters.

In summary, the key contributions of this work are: (1) A diffusion planning framework that models long-horizon plans as chain-structured factor graphs over video segments and enforces boundary agreement on Tweedie estimates rather than on noisy diffusion states. (2) Joint synchronous and asynchronous message passing over denoised variables, coupled with a training-free sampler that guides DDIM steps with diffusion-sphere guidance derived from message-passing residuals, preserving local sample quality, parallelism, and diversity while enforcing boundary agreement. (3) A compositional planning benchmark and empirical study demonstrating significant improvements in temporal coherence, static quality, and task success on held-out start–goal combinations compared with prior compositional baselines (Zhang et al., 2023) that operate on noisy diffusion states.

## 2 RELATED WORK

**Diffusion Models For Planning.** A flurry of work has leveraged diffusion models (Sohl-Dickstein et al., 2015; Ho et al., 2020) for planning (Janner et al., 2022; Ajay et al., 2022; Dong et al., 2024a; He et al., 2023a; Ubukata et al., 2024; Lu et al., 2025; Chen et al., 2024a; Liang et al., 2023; Dong et al., 2024b), with various applications such as path finding (Carvalho et al., 2025; Luo et al., 2024), robotics (Pearce et al., 2023; Fang et al., 2024), and multi-agent (Zhu et al., 2024; Shaoul et al., 2024). Performance can be further improved by increasing test-time compute such as tree search (Feng et al., 2024a; Yoon et al., 2025), hierarchical planning (Li et al., 2023; Chen et al., 2024b), and post-hoc refinement (Lee et al., 2024; Wang et al., 2022). Despite strong results, most prior work studies diffusion planning in low-dimensional state spaces that yield 2D trajectories. While recent work (Xu et al., 2025; Xie et al., 2025; Huang et al., 2024a) begins to consider more complex state spaces, they typically target simple or task-specific scenarios. In this paper, we investigate *visual diffusion planning*, where a plan is represented as a sequence of images, and we introduce a training-free sampling method that scales to significantly longer horizons and to unseen start–goal combinations.

**Compositional Diffusion Generation.** Compositional diffusion models are now well studied (Du et al., 2020; Garipov et al., 2023; Du & Kaelbling, 2024; Mahajan et al., 2024; Okawa et al., 2024; Thornton et al., 2025). One thread develops samplers for logical conjunctions of conditions, combining multiple prompts or constraints into coherent generations (Liu et al., 2022; Bradley et al., 2025; Zhang et al., 2025; Yang et al., 2023). A complementary thread scales the number of inference-time tokens while reusing models trained on short horizons—yielding wide-field panoramas (Zhang et al., 2023; Bar-Tal et al., 2023; Kim et al., 2024a; Lee et al., 2023), longer-duration videos (Wang et al., 2023; Kim et al., 2024b; 2025), and extended-horizon robotic plans (Zhang et al., 2023; Mishra et al., 2023; Luo et al., 2025). However, existing compositional methods suffer from instability

when applied to noisy diffusion states, as they typically rely on score averaging or other heuristic combinations that assume factorization holds throughout the denoising process. In contrast, our approach operates on clean Tweedie estimates rather than noisy intermediate states, formulates the problem as factor graph inference with explicit boundary constraints, and employs principled message passing to maintain global consistency—yielding substantial improvements in both stability and plan quality over prior compositional planning methods.

**Inference-Time Guidance for Diffusion Model.** Inference-time guidance steers diffusion sampling without retraining, enabling adaptive, controllable behavior at test time. This flexibility has driven progress in image restoration (inpainting, deblurring) (Chung et al., 2024; Yang et al., 2024; Yu et al., 2023; Ye et al., 2024; Song et al., 2023a), style transfer (Bansal et al., 2023; He et al., 2023b), and robot motion/behavior generation (Liao et al., 2025; Black et al., 2025; Du & Song, 2025; Song et al., 2023b; Feng et al., 2024b). However, most existing methods steers a fixed-length output, we frame guidance as a form of message passing between tokens—allowing information to propagate across the sequence. This perspective lets us stitch together short behavioral fragments into long-range, temporally consistent visual plan.

## 3 PRELIMINARIES

### 3.1 FACTOR GRAPH FORMULATION FOR COMPOSITIONAL DISTRIBUTIONS

A factor graph $z = [u^1, u^2, \ldots, u^m]$ is a bipartite graph connecting factor nodes $\{x^i\}_{i=1}^n$ and variable nodes $\{u\}_{j=1}^m$, where $x^j \subseteq [u^1, u^2, \ldots, u^m]$. An undirected edge between $x^i$ and $u^j$ exists if and only if $u^j \in x^i$. Given a factor graph that represents the factorization of joint distribution, previous works DiffCollage approximate it with Bethe approximation (Zhang et al., 2023), and Generative Skill Chaining (GSC) extends the same formulation to robot task-and-motion planning as its follow-up work:

$$p(z_t) := \frac{\prod_{i=1}^n p(x_t^i)}{\prod_{j=1}^m p(u_t^j)^{d_j - 1}}, \tag{1}$$

where $d_j$ is the degree of each variable $u_j$. Therefore, the estimated score is base on Bethe approximation is:

$$\nabla_{z_t} \log p(z_t) = \sum_{i=1}^n \nabla_{x_t^i} \log p(x_t^i) + \sum_{j=1}^m (1 - d_j) \nabla_{u_t^j} \log(u_t^j). \tag{2}$$

For example, consider a linear chain $z = [u^1, u^2, u^3, u^4, u^5]$ with factors $x_1 = [u^1, u^2, u^3]$ and $x_2 = [u^3, u^4, u^5]$. We can represent the joint noisy distribution as:

$$p(z_t) = \frac{p(u_t^1, u_t^2, u_t^3)\, p(u_t^3, u_t^4, u_t^5)}{p(u_t^3)}, \tag{3}$$

Given the linear chain graph, the overlapping variable $u^3$ (the one shared between neighboring factors $x_1$ and $x_2$) has degree $d = 2$, while the non-overlapping ones have $d = 1$ (i.e., their distribution come from individual factors). However, Bethe approximation(Eq. 1) holds in clean data (i.e., diffusion timestep $t = 0$), but does not hold when $t > 0$, we further prove its gap later (Theorem 1).

### 3.2 DIFFUSION MODEL AND TRAINING-FREE GUIDED DIFFUSION

Diffusion Models are a class of generative model that generates sample in the desired distribution from an initial Gaussian distribution $p(x_T)$ by iteratively performing a denoising process. It has a pre-defined forward process $q(x_t \mid x_0) = \mathcal{N}(x_t; \sqrt{\bar{\alpha}_t} x_0, (1 - \bar{\alpha}_t)I)$, where $\bar{\alpha}$ is a scalar dependent on diffusion timestep $t$. In this work, we directly estimate Tweedie when training: $\mathbb{E}_{t, x_0, \epsilon_t}[\|x_0 - x_\theta(x_t, t)\|^2]$ ($x_0$ predictor) and apply a DDIM step when sampling:

$$x_{t-1} = \sqrt{\bar{\alpha}_t} x_{0|t} + \sqrt{1 - \bar{\alpha}_t - \sigma_t^2}\, \frac{x_t - \sqrt{\bar{\alpha}_t} x_{0|t}}{\sqrt{1 - \bar{\alpha}_t}} + \sigma_t \epsilon_t, \tag{4}$$

where estimated Tweedie $x_{0|t}$ is the output of $x_\theta(x_t, t)$.

Classifier Guidance (Dhariwal & Nichol, 2021) proposes to train a time dependent classifier in conditional generative tasks. Specifically, the conditional distribution $p(x_t|y)$ can be modeled by

Bayes Rules $p(x_t|y) = p(x_t)p(y|x_t)/p(y) : \nabla_{x_t} \log p(x_t \mid y) = \nabla_{x_t} \log p(x_t) + \nabla_{x_t} \log p(y \mid x_t)$, where $y$ represents the condition or measurement. This paper focuses on conditional guidance in a training-free manner, all the guidance in this paper is in training-free manner. In training-free guidance setting, instead of explicitly training a classifier, the guidance term $p(y \mid x_t)$ can be modeled as a potential function $\exp\left(-L(x_{0|t})\right)$, which simplifies to a gradient-descent update during inference time:

$$\nabla_{x_t} \log p(y \mid x_t) = \nabla_{x_t} \log \frac{\exp\left(-L(x_{0|t})\right)}{Z} = -\nabla_{x_t} L(x_{0|t}), \tag{5}$$

where $x_{0|t} = x_\theta(x_t, t)$ estimated by Tweedie's Predictor. Since $\exp\left(-L(x_{0|t})\right)$ is a point estimation of distribution of $\mathbb{E}_{x_0 \sim p(x_0|x_t)}[\exp(-L(x_0))]$, and the gap between them has a upper bound (Chung et al., 2024) and lower bound (Yang et al., 2024), Diffusion sphere guidance (Yang et al., 2024) is proposed to eliminate this gap based by formulating a constrained optimization problem over a hypersphere with the mean to be $\sqrt{\bar{\alpha}_t}x_{0|t} + \sqrt{1 - \bar{\alpha}_t - \sigma^2}\frac{x_t - \sqrt{\bar{\alpha}_t}x_{0|t}}{\sqrt{1-\bar{\alpha}_t}}$ and radius $\sqrt{s}\sigma_t$ ($s$ is the shape of $x_t$), and derive a closed form solution for the update :

$$x_{t-1} = \sqrt{\bar{\alpha}_t}x_{0|t} + \sqrt{1 - \bar{\alpha}_t - \sigma^2}\frac{x_t - \sqrt{\bar{\alpha}_t}x_{0|t}}{\sqrt{1-\bar{\alpha}_t}} - \sqrt{s}\sigma_t \frac{\nabla_{x_t} L(x_{0|t})}{\left\|\nabla_{x_t} L(x_{0|t})\right\|}. \tag{6}$$

## 4 METHOD

We formulate long-horizon planning as inference over a chain-structured factor graph of overlapping video chunks, where pretrained short-horizon diffusion models provide local priors. Our key innovation is enforcing boundary agreement on estimated clean data (Tweedie estimates) rather than noisy intermediate states, addressing the core limitation that factorization assumptions break down during diffusion sampling. We achieve this through novel synchronous and asynchronous message passing that operates on Tweedie estimates, producing globally consistent guidance without additional training. The approach involves formulating the planning problem as a factor graph (Section 4.1), deriving its distribution (Section 4.2), and sampling via our message passing scheme (Section 4.3).

### 4.1 PROBLEM FORMULATION

While a diffusion model can learn a prior over short, local behaviors, long-horizon planning requires additional structure to ensure feasibility. Beyond satisfying the start and the goal, intermediate pieces must stitch with local consistency. We therefore train a short-horizon diffusion model $x_\theta$ on local task segments; at test time, given a start image and a goal image, we sample from a Gaussian prior, partition the trajectory into overlapping chunks, generate each chunk with $x_\theta$, and compose them into a coherent plan that is finally mapped back to an action sequence through inverse dynamics model.

We represent the plan as a linear chain $z = [u^1, \ldots, u^m]$ and place $n$ overlapping factors $x^i = [u^{2i-1}, u^{2i}, u^{2i+1}]$, $i = 1, \ldots, n$, each collecting three consecutive frames. The endpoints $u^1 = s$ and $u^m = g$ serve as the start and goal boundary variables. Let $A_i$ and $B_i$ denote linear selectors that extract the first and last frames of factor $x^i$, respectively. The feasibility of a plan is enforced by the following boundary agreements:

$$\begin{aligned} \text{(Start/Goal Anchoring)} \quad & A_1 x^1 = s, \qquad B_n x^n = g, \\ \text{(Transition Boundary)} \quad & B_i x^i = A_{i+1} x^{i+1}, \quad i = 1, \ldots, n-1. \end{aligned} \tag{7}$$

This factorization reduces global planning to local generation with explicit boundary equalities and scales by reusing the same local model $x_\theta$ across time while preserving consistency via start–goal anchoring and transition agreements. All factors/variable in latent space encoded by the Cosmos tokenizer (et. al., 2025) into a compact latent representation; we perform planning entirely in this latent space rather than pixels, which reduces dimensionality to save compute.

### 4.2 DISTRIBUTION OF FACTOR GRAPH

Prior work (Zhang et al., 2023; Mishra et al., 2023; 2024) relies on the Bethe-style product of factors normalized by variables (Eq. 1), which is accurate on clean data. Forward diffusion, however, perturbs factorization assumption between factors and variables.

**Theorem 1** (Noisy-Bethe Gap Theorem). *Consider a linear chain* $z = [u^1, u^2, u^3]$ *with pairwise factors* $[u^1, u^2]$ *and* $[u^2, u^3]$*, where* $u^2$ *is the transition boundary variable. Assume the forward noising processes are* $p(u_t^1, u_t^2 \mid u^1, u^2)$, $p(u_t^2, u_t^3 \mid u^2, u^3)$*, and* $p(u_t^2 \mid u^2)$. *Let* $a(u^2) = \int p(u^1, u^2)p(u_t^1, u_t^2 \mid u^1, u^2)\, du^1$, $b(u^2) = \int p(u^2, u^3)p(u_t^2, u_t^3 \mid u^2, u^3)\, du^3$, $c(u^2) = p(u^2)p(u_t^2 \mid u^2)$, $Z = \int c(u^2)\, du^2$*, and* $q(u^2) = c(u^2)/Z$*. Denote by* $p(u_t^1, u_t^2, u_t^3)$ *the true noisy distribution and by* $\hat{p}(u_t^1, u_t^2, u_t^3)$ *the estimator from Eq. 1. Then the gap between true distribution and estimated distribution is:*

$$\Delta = p(u_t^1, u_t^2, u_t^3) - \hat{p}(u_t^1, u_t^2, u_t^3) = Z \operatorname{Cov}_{u^2 \sim q}\left[\frac{a}{c}, \frac{b}{c}\right]. \tag{8}$$

**Interpretation.** The proof is in appendix A. We can view $a(u^2)$ as the left-factor message into the boundary $u^2$, $b(u^2)$ as the right-factor message into the boundary, and $c(u^2)$ as the local boundary evidence. Intuitively, $a(u^2)$ and $b(u^2)$ quantify how the left and right pairwise factors, after passing through their respective forward-noise channels, "vote" for different boundary values $u^2$. The term $c(u^2)$ provides the unary baseline that captures how plausible each boundary value is on its own (and how it transmits noise). The noisy-Bethe gap $\Delta$ is exactly the covariance—under the boundary weighting $q$—between the two relative gains $a(u^2)/c(u^2)$ and $b(u^2)/c(u^2)$. When these gains are uncorrelated (or proportional) across $u^2$, the covariance vanishes and the Bethe approximation remains accurate. Forward diffusion typically introduces shared, heteroscedastic distortions in $u^2$, which make the two gains rise and fall together; this produces a nonzero covariance and, consequently, a systematic gap.

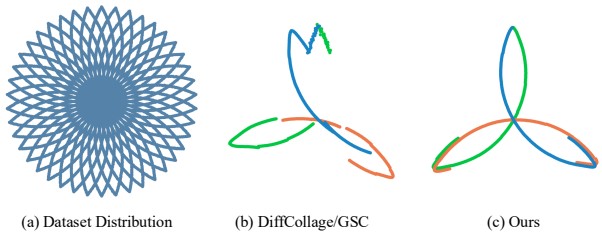

(a) Dataset Distribution     (b) DiffCollage/GSC     (c) Ours

Figure 2: Motivating toy example. We train a short-horizon diffusion model on circular *arc* clips (left). At test time, three $120°$ arc generators are composed to form a three-petal "flower".

Figure 2 illustrates the core failure mode of DiffCollage which is deployed based on noisy factorization assumption. A DiffCollage/GSC-style stitcher (middle) drifts and leaves boundary gaps, while our inference-time message passing (right) aligns shared boundaries and closes the loops. This mirrors the limitation of a Bethe-style product of factors (Eq. 1) under forward diffusion: noise corrupts factorization assumption. Motivated by the Noisy-Bethe gap, instead of enforcing dependencies directly among the noisy factors $x_t^{1:n}$, we impose them on the concatenated Tweedie (denoised) estimates $x_{0|t}^{1:n}$. Accordingly, our approximation to the factor-graph distribution is:

$$p(z_t) = \prod_{i=1}^{n} p(x_t^i) \cdot \exp(-L(x_{0|t}^{1:n})), \tag{9}$$

where $\exp(-L(x_{0|t}^{1:n}))$ acts as a potential that penalizes inconsistencies—and thus enforces dependencies—among the estimated clean variables $x_{0|t}^{1:n} = x_\theta(x_t^{1:n})$.

### 4.3 JOINTLY SYNCHRONOUS AND ASYNCHRONOUS MESSAGE PASSING

Message passing proceeds through *boundary factors*: when the transition boundaries together with the start and goal boundaries agree, the plan is feasible (see Eq. 7). We therefore optimize boundary agreement explicitly. Our synchronous scheme treats the chain as a Gaussian linear system and drives a single residual ($\Sigma^{-1} x_{0|t}^{1:n} = \eta$) to zero via parallel updates, but can be numerically stiff. Our asynchronous scheme uses one-sided, stop-gradient targets to propagate constraints forward and backward in a TD-style manner, yielding faster and more stable convergence at the cost of mild bias. Finally, diffusion-sphere guidance interpolates between unconditional sampling and loss-driven descent, balancing alignment and diversity.

#### 4.3.1 SYNCHRONOUS MESSAGE PASSING

We encode the boundary condition as a Gaussian potential, $\psi_{i-1,i} := \exp(-\frac{1}{c_{i-1}}\|B_{i-1} x_{0|t}^{i-1} - A_i x_{0|t}^i\|^2)$, where $c_{i-1}$ denotes the variance.

**Proposition 1** (Synchronous Message Passing Constraint). *Let $x^{1:n} \in \mathbb{R}^{n \times tchw}$ denote the concatenated intermediate factors in a chain-structured factor graph with transition boundaries $\psi_{i-1,i}$. Given the start boundary $\psi_{s,1}$ and the goal boundary $\psi_{n,g}$, the joint constraints distribution over all intermediate factors is Gaussian:*

$$p_{\text{sync}}(x^{1:n} \,|\, s, g) \;\propto\; \exp(-\tfrac{1}{2}(x^{1:n})^T \Sigma^{-1} x^{1:n} + \eta^T x^{1:n}), \tag{10}$$

*where* $\Sigma^{-1} = \begin{bmatrix} \frac{A_1^T A_1}{c_0} + \frac{B_1^T B_1}{c_1} & -\frac{B_1^T A_2}{c_1} & & \\ -\frac{A_2^T B_1}{c_1} & \frac{A_2^T A_2}{c_1} + \frac{B_2^T B_2}{c_2} & -\frac{B_2^T A_3}{c_2} & \\ & -\frac{A_3^T B_2}{c_2} & \frac{A_3^T A_3}{c_2} + \frac{B_3^T B_3}{c_3} & \\ & & & \ddots \end{bmatrix}, \quad \eta = \begin{bmatrix} \frac{A_1^T s}{c_0} \\ 0 \\ 0 \\ \vdots \\ \frac{B_n^T g}{c_n} \end{bmatrix}.$

The detailed proof is in appendix B. We perform synchronous message passing on the estimated Tweedie factors $x_{0|t}^{1:n}$ by penalizing the deviation from the consistent linear system $\Sigma^{-1} x_{0|t}^{1:n} = \eta$. In practice, we set $c_i = 1$ for all $i = 0, \dots, n.$:

$$L_{sync} = \left\| \Sigma^{-1} x_{0|t}^{1:n} - \eta \right\| \tag{11}$$

Here, synchronous refers to a lockstep update schedule, in which all updates are computed from the same current iterate and applied simultaneously. This scheme preserves parallelism, eliminates order-dependent effects, and guides the Tweedie estimates toward satisfying the chain constraints.

### 4.3.2 ASYNCHRONOUS MESSAGE PASSING

While the synchronous objective is conceptually clean, the resulting hard consistency constraint is difficult to optimize and often exhibits slow or unstable convergence (Ortiz et al., 2021). To improve stability and speed, we adopt an asynchronous scheme with bootstrapped targets and stop-gradient, akin to temporal-difference updates Hansen et al. (2024); Li et al. (2025). Concretely, we optimize:

$$
L_{async} = \underbrace{\left\| s - A_1 x_{0|t}^1 \right\| + \sum_{i=1}^{n-1} \gamma^i \left\| sg(B_i \hat{x}_{0|t}^i) - A_{i+1} x_{0|t}^{i+1} \right\|}_{\text{forward passing}}
$$
$$
+ \underbrace{\sum_{i=1}^{n-1} \gamma^{n-i} \left\| B_i x_{0|t}^i - sg(A_{i+1} \hat{x}_{0|t}^{i+1}) \right\| + \left\| B_n x_{0|t}^n - g \right\|}_{\text{backward passing}}, \tag{12}
$$

where $sg(\cdot)$ is the stop-gradient operator, $\hat{x}_{0|t}^{i+1}$ is a target produced by diffusion model with latest parameters, and $x_{0|t}^i$ is produced by an EMA of the model parameters. The discount $\gamma$ down-weights messages as they move away from the start or the goal.

The boundary terms $\left\| s - A_1 x_{0|t}^1 \right\|$ and $\left\| B_n x_{0|t}^n - g \right\|$ anchor the chain to the start and goal. The forward message loss penalizes mismatch between $B_i \hat{x}_{0|t}^i$ (outgoing) and $A_{i+1} x_{0|t}^{i+1}$ (incoming), with $sg(\cdot)$ enforcing one-way forward passing. Similarly, the backward message loss mirrors the same constraint in the reverse direction.

### 4.3.3 DIFFUSION-SPHERE GUIDED MESSAGE PASSING

Having derived differentiable losses for synchronous and asynchronous message passing, we adopt the training-free guidance of DSG (Yang et al., 2024). As noted by DSG (Eq. 6), stronger guidance improves alignment but can reduce sample diversity. To balance alignment and exploration, we interpolate between the unconditional sampling direction and the normalized descent direction induced by our loss:

$$d_m = d^{sample} + g_r(d^* - d^{sample}), \quad x_{t-1}^{1:n} = \mu_{t-1}^{1:n} + r \frac{d_m}{\|d_m\|}. \tag{13}$$

---

**Algorithm 1** Compositional Generation

---

1: **Require:** Model EMA $x_\theta$, Latest Model $\hat{x}_\theta$
2: **Hyperparameters:** Diffusion Time Step $T$, number of factors chained $n$, guidance weight $g$
3: Sample $z_T \sim \mathcal{N}(0, I)$
4: Split $z_T$ to $n$ overlapping chunks $x_T^{1:n}$
5: **for** $t = T$ to $1$ **do**
6:    $x_{0|t}^{1:n} = x_\theta(x_t^{1:n})$                                          ▷ forward model passing
7:    $\hat{x}_{0|t}^{1:n} = \hat{x}_\theta(x_t^{1:n})$
8:    $\mu_{t-1}^{1:n} = \sqrt{\bar{\alpha}_t} x_{0|t}^{1:n} + \sqrt{1 - \bar{\alpha}_t - \sigma^2} \frac{x_t^{1:n} - \sqrt{\bar{\alpha}_t} x_{0|t}^{1:n}}{\sqrt{1 - \bar{\alpha}_t}}$            ▷ DDIM Step
9:    $L = L_{sync} + L_{async}$           ▷ Jointly sync and async message passing
10:    $d^* = -\sqrt{s}\sigma_t \cdot \frac{\nabla_{x_t^{1:n}} L}{\|L\|}$            ▷ Diffusion Sphere Guidance
11:    $d^{sample} = \sigma_t \epsilon_t$
12:    $d_m = d^{sample} + g(d^* - d^{sample})$
13:    $x_{t-1}^{1:n} = \mu_{t-1}^{1:n} + r \frac{d_m}{\|d_m\|}$
14: **end for**
15: merge chunks $x_0^{1:n}$ to get final plan $z_0$
16: **return** $z_0$

---

Here $d^{sample} = \sigma_t \epsilon_t$ is the unconditional annealing step, $d^* = -\sqrt{s}\,\sigma_t \cdot \frac{\nabla_{x_t^{1:n}} L}{\|\nabla_{x_t^{1:n}} L\|}$ is the steepest descent direction for our sync/async objective, and $d_m$ is subsequently normalized to satisfy the spherical-Gaussian constraint.

### 4.4 Compositional Video Planning via Inference-Time Diffusion Scaling

The procedure for compositional generation is summarized in Algorithm 1. At a high level, the goal is to generate long-horizon trajectories by decomposing them into overlapping local factors and enforcing boundary agreement across those factors during the diffusion sampling process. At each timestep, DDIM provides the base update, while a joint synchronous–asynchronous message passing loss defines a residual that Diffusion Sphere Guidance interpolates against, steering updates toward agreement without collapsing diversity. The resulting updates are local and parallel across overlapping factors yet collectively converge to a feasible, consistent plan. After all steps, merging the denoised chunks yields a smooth, temporally aligned trajectory $z_0$.

For robot manipulation planning, we train a video diffusion model on randomly sampled short chunks from long-horizon demonstrations and an inverse dynamics model that predicts actions from consecutive frames. At test time, we condition the diffusion model on start and goal images and apply the compositional generation procedure to produce complete video plans. The resulting visual trajectory is converted into executable robot actions via the inverse dynamics model. The procedure is training-free, plug-and-play, and compatible with unconditional short-horizon diffusion backbones, enabling generalization to unseen start-goal combinations without task-specific retraining.

## 5 Experiments

We present experiment results on multiple robotic manipulation scenes spanning 100 tasks (18 in distribution, 82 out of distribution) of varying difficulties. Our objective is to investigate (1) the visual fidelity of the generated video plans (Section 5.1) (2) how the proposed compositional visual planning can generalize to long-horizon unseen tasks (Section 5.2) (3) how each proposed component affect the generation performance of our method (Section 5.3) (4) the effectiveness of our method in real robot manipulation tasks (Section 5.4).

**Compositional Planning Benchmark.** We develop a simulation benchmark for compositional planning in robotic manipulation based on ManiSkill (Mu et al., 2021), where each scene contains $N$ start states and $N$ goal states, resulting in $N \cdot N$ tasks (i.e., different start/goal pairs) per scene, as shown in Appendix E.

Our training datasets only contains demonstrations for $N$ start-goal pairs. At test time, we evaluate the planner on both the $N$ seen start–goal pairs (in-distribution) and the remaining $N \cdot N - N$

unseen pairs (out-of-distribution), since a capable planner should generalize to new combinations if the dataset covers all the required fragments. For example, in the Real World setting (Figure 3), demonstrations cover only the orange or blue regions; the planner must generalize across them to form cross-region plans unseen in the dataset but composable from its fragments. We address this type of generation by learning from short demonstration chunks randomly taken from long-horizon tasks and compositionally generates multiple chunks at inference time to construct the final plan.

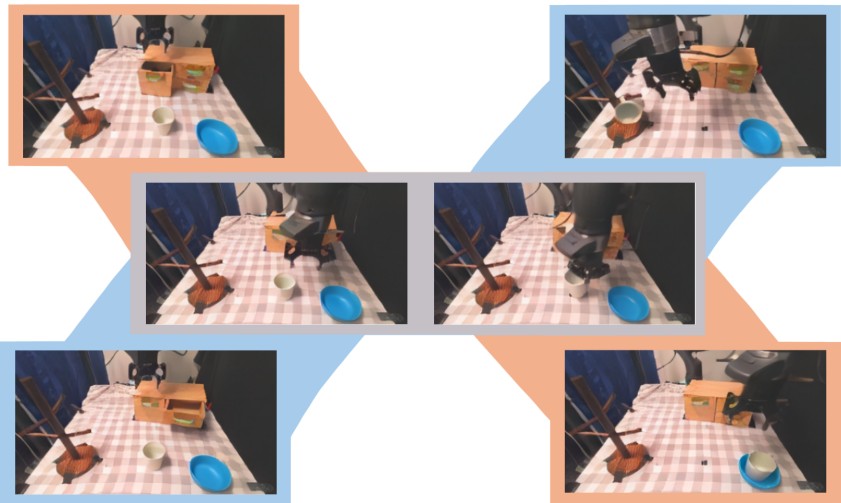

Figure 3: **Real World Task Layout.** This task involves 2 start and 2 goal configurations. We also evaluate on more challenging tasks. A complete list of task settings is provided in Appendix E.

**Evaluation Setup.** Given specified start and goal image as task context, our method first synthesize a sequence of frames as subgoals. We then use an MLP-based inverse dynamics model to predict the pose of end-effector for the robot to execute, conditioned on adjacent images. The inverse dynamics models are trained using the same demonstrations as the planner conditioned on adjacent images. We report both the video quality metrics and the success rates of the planners over the robotic manipulation tasks. An episode is counted as success if the target objects ends up the specified state within a small tolerance. For each environment, we report the success rate over all evaluation episodes. We evaluate all methods with 5 random seeds for each experiment and report the mean and standard deviation.

**Baselines.** For both policy-based and composition-based baseline, details are in Appendix D. For compositional baseline **DiffCollage/GSC** (where GSC is DiffCollage adapted to robotic planning), we condition on the start and goal images to generate an entire plan by noisy factorization Eq. 2, and then use an inverse dynamics model to execute that plan.

## 5.1 Video Generation Quality Study

Beyond reporting task success rate, we also evaluate the visual fidelity of the synthesized video plans. Even when rollouts verify feasibility, perceptual quality still warrants careful analysis. Accordingly, we score generated videos with VBench++ (Huang et al., 2024b), focusing on robotics-centric metrics that matter for control: *Dynamic Quality* (inter-frame), which includes (i) motion smoothness—capturing temporal stability of robot/object motion—and (ii) background consistency—testing whether the scene remains coherent over time; and *Static Quality* (frame-wise), which includes Aesthetic and Imaging Quality to ensure frames are clear and largely artifact-free. Our generation strategy substantially improves the time-dependent properties that matter for control: across all scenes and distributions, Motion Smoothness and Background Consistency far exceed DiffCollage (Zhang et al., 2023). This translates into dynamically executable trajectories and coherent spatiotemporal scenes. In terms of static quality, Aesthetic remains comparable, while Imaging shows a consistent and large advantage, directly reflecting fewer blurry frames and cleaner visuals.

| Scene | Type | Dynamic Quality ↑ | | | | Static Quality ↑ | | | |
|---|---|---|---|---|---|---|---|---|---|
| | | Motion Smoothness | | Background Consistency | | Aesthetic | | Imaging | |
| | | DiffCollage | Ours | DiffCollage | Ours | DiffCollage | Ours | DiffCollage | Ours |
| Overall | **IND** | $0.88_{\pm0.04}$ | $\mathbf{0.94}_{\pm0.06}$ | $0.81_{\pm0.04}$ | $\mathbf{0.90}_{\pm0.04}$ | $0.49_{\pm0.03}$ | $\mathbf{0.50}_{\pm0.03}$ | $0.60_{\pm0.05}$ | $\mathbf{0.70}_{\pm0.03}$ |
| | **OOD** | $0.87_{\pm0.06}$ | $\mathbf{0.97}_{\pm0.05}$ | $0.80_{\pm0.07}$ | $\mathbf{0.90}_{\pm0.05}$ | $0.46_{\pm0.04}$ | $\mathbf{0.48}_{\pm0.03}$ | $0.55_{\pm0.05}$ | $\mathbf{0.69}_{\pm0.05}$ |

Table 1: **Comparison across four scenes on Dynamic/Static Quality.** Our results are averaged over 5 seeds and standard deviations are shown after the ± sign.

## 5.2 COMPOSITIONAL PLANNING BENCHMARK

We present the robot manipulation success rates of 4 different escenes in Table 2. We separately report the success rates of IND and OOD tasks, where IND represents the $N$ tasks seen in the training data while OOD represents the $N \cdot N - N$ unseen tasks. We observe that DiffCollage fails at almost all tasks. Qualitatively, we find that the synthesized images of DiffCollage tend to be blurry or even unrealisitc, perhaps due to its score averaging sampling scheme. Such suboptimal images will further confuses the inverse dynamic models, cause unstable behaviors and failures. In contrast, our method achieves significantly higher success rates, indicating that the generated visual plans are realistic and accurate for the inverse dynamic model to follow. We also include several multiple representative policy learning baselines, such as Goal-Conditioned Diffusion Policy (GCDP). We notice that though strong policy learning baseline is able to perform well on IND tasks, their performance suffers from a significant degradation on OOD tasks. In contrast, our method—enabled by the graphical chain formulation and message passing—maintains stable performance regardless of the task distribution.

| Scene | Type | LCBC | LCDP | GCBC | GCDP | DiffCollage | CompDiffuser | Ours |
|---|---|---|---|---|---|---|---|---|
| Tool-Use | **IND** | $80_{\pm7}$ | $95_{\pm2}$ | $85_{\pm5}$ | $96_{\pm3}$ | $1_{\pm2}$ | $60_{\pm2}$ | $\mathbf{97}_{\pm3}$ |
| | **OOD** | $15_{\pm3}$ | $37_{\pm8}$ | $13_{\pm6}$ | $42_{\pm13}$ | $0_{\pm0}$ | $51_{\pm3}$ | $\mathbf{96}_{\pm2}$ |
| Drawer | **IND** | $35_{\pm6}$ | $54_{\pm6}$ | $30_{\pm5}$ | $50_{\pm6}$ | $0_{\pm0}$ | $20_{\pm5}$ | $\mathbf{53}_{\pm5}$ |
| | **OOD** | $6_{\pm5}$ | $26_{\pm14}$ | $7_{\pm5}$ | $18_{\pm16}$ | $0_{\pm0}$ | $18_{\pm3}$ | $\mathbf{52}_{\pm6}$ |
| Cube | **IND** | $28_{\pm3}$ | $58_{\pm5}$ | $26_{\pm4}$ | $60_{\pm3}$ | $0_{\pm0}$ | $32_{\pm8}$ | $\mathbf{64}_{\pm10}$ |
| | **OOD** | $8_{\pm3}$ | $22_{\pm12}$ | $5_{\pm5}$ | $24_{\pm13}$ | $0_{\pm0}$ | $34_{\pm6}$ | $\mathbf{65}_{\pm9}$ |
| Puzzle | **IND** | $23_{\pm5}$ | $48_{\pm5}$ | $19_{\pm6}$ | $47_{\pm3}$ | $0_{\pm0}$ | $10_{\pm3}$ | $\mathbf{50}_{\pm11}$ |
| | **OOD** | $0_{\pm0}$ | $11_{\pm9}$ | $0_{\pm0}$ | $12_{\pm11}$ | $0_{\pm0}$ | $9_{\pm3}$ | $\mathbf{50}_{\pm13}$ |
| Overall | **IND** | $33_{\pm18}$ | $57_{\pm15}$ | $30_{\pm21}$ | $56_{\pm16}$ | $0_{\pm1}$ | $17_{\pm2}$ | $\mathbf{59}_{\pm17}$ |
| | **OOD** | $2_{\pm4}$ | $15_{\pm12}$ | $15_{\pm12}$ | $15_{\pm13}$ | $0_{\pm0}$ | $16_{\pm2}$ | $\mathbf{54}_{\pm14}$ |

Table 2: **Quantitative Results on Compositional Planning Bench.** We benchmark our method on the 100 test-time tasks across 4 scenes with 30 episodes per task. Our results are averaged over 5 seeds and standard deviations are shown after the ± sign.

## 5.3 ABLATION STUDIES

**Jointly Synchronous & Asynchronous Message Passing.** We compare the success rates of three variants of our test-time compositional sampling scheme in Cube Scene: Only Synchronous Loss (*Sync Only*), Only Asynchronous Loss (*Async Only*), and Joint Synchronous and Asynchronous Loss (*Sync & Async*), as shown in Figure 4. *Sync only* suffers from overly tight constraints that are difficult to optimize, leading to lower success rates. In contrast, the asynchronous variant performs better. Combining the two—*Sync & Async*—outperforms either alone, likely due to its more effective balance of constraint enforcement and flexibility.

**Scaling in Sampling Steps.** We study how the number of diffusion sampling steps affects the planning performance on Drawer Scene (Figure 5). Success rates improve as the number of steps increases, demonstrating that our method scales effectively with additional test-time compute. We hypothesize that taking more steps enables deeper cross-factor message passing through repeated

denoising and guidance updates, which in turn reduces boundary inconsistencies and yields more accurate, temporally coherent plans.

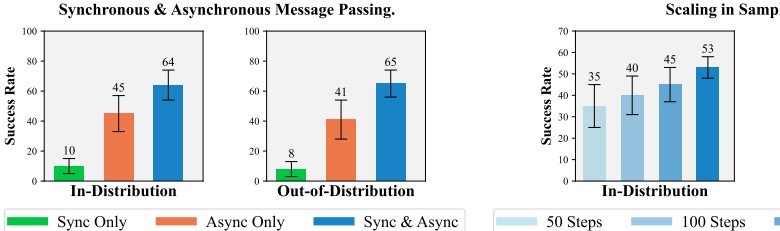

Figure 4: **Effect of synchronous and asynchronous message passing.**

Figure 5: **Effect of sampling steps on planning performance.**

### 5.4 REAL ROBOT EXPERIMENT

For the real-world experiments, we deploy our method on a Franka Emika Panda robot, controlled at 20Hz using joint impedance control. Visual image observations are captured using an Intel RealSense D435 depth camera. For data collection, the robot is teleoperated using a Meta Quest 3 headset, with tracked Cartesian poses converted to joint configurations through inverse kinematics.

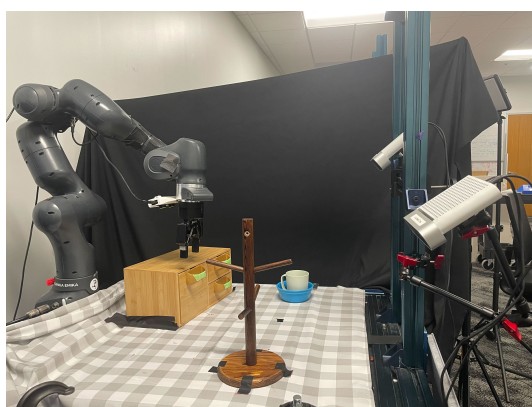

Figure 6: **Hardware Setup.** We deploy our method on a Franka Emika Panda robot.

| Real Scene | IND:Task1 | IND:Task2 | OOD:Task3 | OOD:Task4 |
|---|---|---|---|---|
| DiffCollage | 1/10 | 1/10 | 0/10 | 0/10 |
| Ours | 9/10 | 7/10 | 10/10 | 8/10 |

Table 3: **Real-robot success rates.** Our method substantially outperforms DiffCollage across both in-distribution (IND) and out-of-distribution (OOD) tasks on real hardware.

## 6 CONCLUSION

We introduced Compositional Visual Planning, an inference-time method that composes long-horizon plans by stitching overlapping video factors with message passing on Tweedie estimates. A chain-structured factor graph imposes global consistency, enforced via joint synchronous and asynchronous updates, while diffusion-sphere guidance balances alignment and diversity without retraining. Compositional Visual Planning is plug-and-play with short-horizon diffusion video prediction model, scales with test-time compute, and generalizes to unseen start–goal combinations. Beyond robotics, the framework is applicable to broader domains, such as panorama image generation and long-form text-to-video synthesis, which we leave for future exploration.

## REPRODUCIBILITY STATEMENT

We promise to provide all the source code to reproduce the results in this paper, including the proposed algorithm and the evaluation benchmark.

## ACKNOWLEDGMENTS

This work was partially supported by the National Science Foundation under Awards NSF-2442393, NSF-2409016, and NSF-1942523, as well as by Samsung Research America. The views and conclusions expressed in this material are those of the authors and do not necessarily reflect the official policies or endorsements of the funding agencies. We would like to thank Shangzhe Li for helpful early discussion on trajectory stitching and feedback of the manuscript.

## LLM USAGE DISCLOSURE

We confirm that no large language models (LLMs) were used to generate, edit, or refine the scientific content, experimental design, results, or conclusions of this paper. All text, figures, algorithms, and analyses were produced entirely by the authors.

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

APPENDIX

## A NOISY-BETHE GAP THEOREM

**Theorem 1** (Noisy-Bethe Gap Theorem). *Consider a linear chain $z = [u^1, u^2, u^3]$ with pairwise factors $[u^1, u^2]$ and $[u^2, u^3]$, where $u^2$ is the transition boundary variable. Assume the forward noising processes are $p(u_t^1, u_t^2 \,|\, u^1, u^2)$, $p(u_t^2, u_t^3 \,|\, u^2, u^3)$, and $p(u_t^2 \,|\, u^2)$. Let $a(u^2) = \int p(u^1, u^2)p(u_t^1, u_t^2 \,|\, u^1, u^2)\, du^1$, $b(u^2) = \int p(u^2, u^3)p(u_t^2, u_t^3 \,|\, u^2, u^3)\, du^3$, $c(u^2) = p(u^2)p(u_t^2 \,|\, u^2)$, $Z = \int c(u^2)\, du^2$, and $q(u^2) = c(u^2)/Z$. Denote by $p(u_t^1, u_t^2, u_t^3)$ the true noisy distribution and by $\hat{p}(u_t^1, u_t^2, u_t^3)$ the estimator from Eq. 1. Then the gap between true distribution and estimated distribution is:*

$$\Delta = p(u_t^1, u_t^2, u_t^3) - \hat{p}(u_t^1, u_t^2, u_t^3) = Z\,\mathrm{Cov}_{u^2 \sim q}\left[\tfrac{a}{c},\ \tfrac{b}{c}\right]. \tag{8}$$

*Proof.* The true noisy distribution is:

$$
\begin{aligned}
p(u_t^1, u_t^2, u_t^3) &= \int p(u_t^1, u_t^2, u_t^3, u^1, u^2, u^3)\, du^1 du^2 du^3 \\
&= \int p(u^1, u^2, u^3)p(u_t^1, u_t^2, u_t^3 | u^1, u^2, u^3)\, du^1 du^2 du^3 \\
&= \int \frac{p(u^1, u^2)p(u_t^1, u_t^2 \,|\, u^1, u^2)p(u^2, u^3)p(u_t^2, u_t^3 \,|\, u^2, u^3)}{p(u^2)p(u_t^2 | u^2)}\, du^1 du^2 du^3 \\
&= \int \frac{\int p(u^1, u^2)p(u_t^1, u_t^2 \,|\, u^1, u^2)du^1 \int p(u^2, u^3)p(u_t^2, u_t^3 \,|\, u^2, u^3)du^3}{p(u^2)p(u_t^2 | u^2)}\, du^2
\end{aligned}
\tag{14}
$$

The estimator used in (Zhang et al., 2023; Mishra et al., 2023; 2024) is:

$$\hat{p}(u_t^1, u_t^2, u_t^3) = \frac{\int p(u^1, u^2)p(u_t^1, u_t^2 \,|\, u^1, u^2)\, du^1 du^2 \int p(u^2, u^3)p(u_t^2, u_t^3 \,|\, u^2, u^3)\, du^2 du^3}{\int p(u^2)p(u_t^2 \,|\, u^2)\, du^2} \tag{15}$$

Define the left-factor message into $u^2$ as $a(u^2) = \int p(u^1, u^2)p(u_t^1, u_t^2 \,|\, u^1, u^2)\, du^1$, the right-factor message into $u^2$ as $b(u^2) = \int p(u^2, u^3)p(u_t^2, u_t^3 \,|\, u^2, u^3)\, du^3$, and the local boundary evidence $c(u^2) = p(u^2)p(u_t^2 \,|\, u^2)$. Then $p(u_t^1, u_t^2, u_t^3) = \int \frac{ab}{c}du^2$ and $\hat{p}(u_t^1, u_t^2, u_t^3) = \frac{\int a\, du^2 \int b\, du^2}{\int c\, du^2}$.

Introduce a change of measure by setting $q(u^2) = \frac{c}{Z}$ with $Z = \int c\, du^2$. For the true distribution, we have:

$$
\begin{aligned}
&p(u_t^1, u_t^2, u_t^3) \\
&= \int \frac{\int p(u^1, u^2)p(u_t^1, u_t^2 \,|\, u^1, u^2)du^1 \int p(u^2, u^3)p(u_t^2, u_t^3 \,|\, u^2, u^3)du^3}{p(u^2)p(u_t^2|u^2)} \frac{p(u^2)p(u_t^2|u^2)}{p(u^2)p(u_t^2|u^2)} \frac{Z}{Z}\, du^2 \\
&= Z \int \frac{\int p(u^1, u^2)p(u_t^1, u_t^2 \,|\, u^1, u^2)du^1}{p(u^2)p(u_t^2|u^2)} \frac{\int p(u^2, u^3)p(u_t^2, u_t^3 \,|\, u^2, u^3)du^3}{p(u^2)p(u_t^2|u^2)} \frac{p(u^2)p(u_t^2|u^2)}{Z}du^2 \\
&= Z\,\mathbb{E}_{u \sim q(u^2)}[\tfrac{a}{c}\tfrac{b}{c}]
\end{aligned}
\tag{16}
$$

For the estimator, observe that:

$$
\begin{aligned}
\int a(u^2)\, du^2 &= \int \left( \int p(u^1, u^2)p(u_t^1, u_t^2 \,|\, u^1, u^2)\, du^1 \right) du^2 \\
&= \int \left( \int p(u^1, u^2)p(u_t^1, u_t^2 \,|\, u^1, u^2)\, du^1 \right) \frac{p(u_2)p(u_t|u_2)}{p(u_2)p(u_t|u_2)} \frac{Z}{Z}du^2 \\
&= Z \int \frac{\left( \int p(u^1, u^2)p(u_t^1, u_t^2 \,|\, u^1, u^2)\, du^1 \right)}{p(u_2)p(u_t|u_2)} \frac{p(u_2)p(u_t|u_2)}{Z}du^2 \\
&= Z\,\mathbb{E}_{u^2 \sim q(u^2)}[\tfrac{a}{c}]
\end{aligned}
\tag{17}
$$

By the same argument, $\int b(u^2)du^2 = Z\,\mathbb{E}_{u^2\sim q(u^2)}[\frac{b}{c}]$, therefore the estimated distribution:

$$\hat{p}(u_t^1, u_t^2, u_t^3) = Z\,\mathbb{E}_{u^2\sim q(u^2)}[\tfrac{a}{c}]\mathbb{E}_{u^2\sim q(u^2)}[\tfrac{b}{c}]. \tag{18}$$

Finally, the difference between the true and estimated distributions is:

$$\begin{aligned}
\Delta &= p(u_t^1, u_t^2, u_t^3) - \hat{p}(u_t^1, u_t^2, u_t^3)\\
&= Z\,\mathbb{E}_{u\sim q(u^2)}[\tfrac{a}{c}\tfrac{b}{c}] - Z\,\mathbb{E}_{u^2\sim q(u^2)}[\tfrac{a}{c}]\mathbb{E}_{u^2\sim q(u^2)}[\tfrac{b}{c}]\\
&= Z\,\mathrm{Cov}[\tfrac{a}{c}, \tfrac{b}{c}].
\end{aligned} \tag{19}$$

This shows that the estimator departs from the true distribution by a covariance term under the reweighted boundary measure $q(u^2)$, scaled by $Z$. $\qquad\square$

## B  SYNCHRONOUS MESSAGE PASSING

**Proposition 1** (Synchronous Message Passing Constraint). *Let $x^{1:n} \in \mathbb{R}^{n\times tchw}$ denote the concatenated intermediate factors in a chain-structured factor graph with transition boundaries $\psi_{i-1,i}$. Given the start boundary $\psi_{s,1}$ and the goal boundary $\psi_{n,g}$, the joint constraints distribution over all intermediate factors is Gaussian:*

$$p_{\text{sync}}(x^{1:n}\,|\,s,\,g) \;\propto\; \exp(-\tfrac{1}{2}(x^{1:n})^T\Sigma^{-1}x^{1:n} + \eta^T x^{1:n}), \tag{10}$$

*where* $\Sigma^{-1} = \begin{bmatrix} \frac{A_1^TA_1}{c_0}+\frac{B_1^TB_1}{c_1} & -\frac{B_1^TA_2}{c_1} & & \\ -\frac{A_2^TB_1}{c_1} & \frac{A_2^TA_2}{c_1}+\frac{B_2^TB_2}{c_2} & -\frac{B_2^TA_3}{c_2} & \\ & -\frac{A_3^TB_2}{c_2} & \frac{A_3^TA_3}{c_2}+\frac{B_3^TB_3}{c_3} & \\ & & & \ddots \end{bmatrix}, \quad \eta = \begin{bmatrix} \frac{A_1^Ts}{c_0} \\ 0 \\ 0 \\ \vdots \\ \frac{B_n^Tg}{c_n} \end{bmatrix}.$

*Proof.* $p_{sync}(x_{1:n}\,|\,s,\,g)$ can be represented as a product of dependencies over all boundary variables:

$$p(x^{1:n}\,|\,s,\,g)_{sync} \;=\; \psi_{s,1}(s,x_1)\,\psi_{1,2}(x_1,x_2)\,\cdots\,\psi_{n-1}(x_{n-1},x_n)\,\psi_{n,g}(x_n,g). \tag{20}$$

The aim of this equation is to express the joint distribution over all intermediate states, given initial and final state:

$$\begin{aligned}
p(x^{1:n}\,|\,s,\,g) &= \psi_0(s,x_1)\,\psi_1(x_1,x_2)\,\cdots\,\psi_{n-1}(x_{n-1},x_n)\,\psi_n(x_n,g)\\[4pt]
&\propto \exp\Big(-\frac{1}{c_0}\|s-A_1x_1\|^2 - \frac{1}{c_1}\|B_1x_1-A_2x_2\|^2 - \frac{1}{c_2}\|B_2x_2-A_3x_3\|^2\\
&\qquad - \cdots - \frac{1}{c_{n-1}}\|B_{n-1}x_{n-1}-A_nx_n\|^2 - \frac{1}{c_n}\|B_nx_n-g\|^2\Big)\\[4pt]
&= \exp\Big(-\tfrac{1}{2}s^T\frac{I}{c_0}s + \tfrac{1}{2}s^T\frac{A_1}{c_0}x_1 + \tfrac{1}{2}x_1^T\frac{A_1}{c_0}s - \tfrac{1}{2}x_1^T\frac{A_1^TA_1}{c_0}x_1\\
&\qquad -\tfrac{1}{2}x_1^T\frac{B_1^TB_1}{c_1}x_1 + \tfrac{1}{2}x_1^T\frac{B_1^TA_2}{c_1}x_2 + \tfrac{1}{2}x_2^T\frac{A_2^TB_1}{c_1}x_1 - \tfrac{1}{2}x_2^T\frac{A_2^TA_2}{c_1}x_2\\
&\qquad -\tfrac{1}{2}x_2^T\frac{B_2^TB_2}{c_2}x_2 + \tfrac{1}{2}x_2^T\frac{B_2^TA_3}{c_2}x_3 + \tfrac{1}{2}x_3^T\frac{A_3^TB_2}{c_2}x_2 - \tfrac{1}{2}x_3^T\frac{A_3^TA_3}{c_2}x_3\\
&\qquad -\tfrac{1}{2}x_3^T\frac{B_3^TB_3}{c_3}x_3 + \tfrac{1}{2}x_3^T\frac{B_3^TA_4}{c_3}x_4 + \tfrac{1}{2}x_4^T\frac{A_4^TB_3}{c_3}x_3 - \tfrac{1}{2}x_4^T\frac{A_4^TA_4}{c_3}x_4\\
&\qquad\qquad\qquad\qquad\vdots\\
&\qquad -\tfrac{1}{2}x_n^T\frac{B_n^TB_n}{c_n}x_n + \tfrac{1}{2}x_n^T\frac{B_n^T}{c_n}g + \tfrac{1}{2}g^T\frac{B_n}{c_n}x_n - \tfrac{1}{2}g^T\frac{I}{c_n}g\Big)\\[4pt]
&= \exp\big(-\frac{1}{2}x_{1:n}^T\Sigma^{-1}x_{1:n} + \eta^T x_{1:n}\big),
\end{aligned} \tag{21}$$

where $\Sigma^{-1} = \begin{bmatrix} \frac{A_1^T A_1}{c_0} + \frac{B_1^T B_1}{c_1} & -\frac{B_1^T A_2}{c_1} & & \\ -\frac{A_2^T B_1}{c_1} & \frac{A_2^T A_2}{c_1} + \frac{B_2^T B_2}{c_2} & -\frac{B_2^T A_3}{c_2} & \\ & -\frac{A_3^T B_2}{c_2} & \frac{A_3^T A_3}{c_2} + \frac{B_3^T B_3}{c_3} & \\ & & & \ddots \end{bmatrix}, \eta = \begin{bmatrix} \frac{A_1^T s}{c_0} \\ 0 \\ 0 \\ \vdots \\ \frac{B_n^T g}{c_n} \end{bmatrix}.$

## C  LIMITATIONS

Our method has several limitations. First, it relies on the accuracy of the estimated clean data (Tweedie estimates) during denoising, since guidance losses are computed on these estimates. This sensitivity could be mitigated by performing multi-step Tweedie estimation or by scaling up training data and model capacity. Second, as in prior work, the number of test-time composed segments, $n$, must be specified manually. Developing procedures that automatically infer $n$ from task structure and uncertainty would be an interesting future research direction. Lastly, our approach can be more computationally demanding than direct averaging-based sampling, because test-time guidance is implemented via gradient-based optimization(Table 5). Exploring lighter optimization schedules could potentially reduce this overhead.

## D  BASELINE IMPLEMENTATIONS

To ensure clarity about how the prior baselines are instantiated in our setting, we briefly summarize the exact formulations of all baselines below. We'd like to clarify DiffCollage / GSC/ CompDiffuser are joint denoising over the full long-horizon trajectory, starting from a single long-horizon Gaussian.

**Language-Conditioned Behavioral Cloning (LCBC).** The policy uses a T5 text encoder to embed natural-language instructions. We concatenate the text embedding with image features extracted by a ResNet backbone, and feed the result into an MLP policy head. The model is trained in a supervised manner to predict a single action at each timestep, conditioned on both the language input and the current observation.

**Language-Conditioned Diffusion Policy (LCDP).** LCDP follows the same text encoding pipeline as LCBC with a T5 encoder, but replaces the MLP head with a Transformer-based policy head. The Transformer generates chunks of actions rather than single-step predictions, allowing multi-step reasoning conditioned on language and observations.

**Goal-Conditioned Behavioral Cloning (GCBC).** GCBC uses a ResNet backbone to encode both the current observation image and the goal image. The concatenated features are passed through an MLP policy head, which outputs a single action. This provides a goal-aware baseline without language conditioning.

**Goal-Conditioned Diffusion Policy (GCDP).** GCDP employs a ResNet backbone with a Transformer policy head, conditioned jointly on the current observation and the goal image. The model outputs action chunks, enabling multi-step planning toward the goal state.

**DiffCollage / GSC.** DiffCollage is a compositional test time generation method, and GSC refers to its adaptation for robotic planning. Given a start image and a goal image, the model directly generates an entire visual plan (a sequence of intermediate frames). It composes a long-horizon score following Eq. 2 and iteratively denoise the entire long trajectory. We then employ a separately trained inverse dynamics model to convert the visual plan into executable robot actions.

**CompDiffuser.** CompDiffuser is a trained joint-denoising baseline: it trains the short-horizon model to condition on its preceding and following noisy chunks. At test time, it subdivides the long trajectory into overlapping chunks, and denoises them jointly while conditioning each chunk on its neighbors. We then employ a separately trained inverse dynamics model to convert the visual plan into executable robot actions. CompDiffuser is SOTA joint-denoising baseline on OGBench 2D xy point maze navigation; however, its behavior on high-dimensional visual planning had not been evaluated prior to our experiments.

# E    COMPOSITIONAL PLANNING TASKS

## E.1    DATASET AND SIMULATION SETUP

**Assets.** Our asset library combines 3D models and textures from ShapeNet (Chang et al., 2015) and RoboCasa (Nasiriany et al., 2024). We additionally apply simple high-quality texture(e.g., wood, plastic, metal finishes) to increase visual fidelity. All simulations are conducted in the SAPIEN engine (Xiang et al., 2020), which provides high-fidelity physics and rendering for robotic manipulation.

**State and Action Space.** The observation space consists solely of RGB images with resolution $256 \times 256 \times 3$, without access to privileged information such as depth or ground-truth states. The action space is parameterized by the end-effector position (3D Cartesian coordinates), orientation represented as a quaternion (4D), and a binary scalar controlling the gripper open/close state. During initialization, all object poses are randomized within a $0.2\,\mathrm{m}$ radius from their nominal positions, ensuring sufficient variability and out-of-distribution test cases.

**Demonstrations.** We provide 300 expert demonstrations for each of the $N$ start–goal combinations across all scenes, resulting in thousands of trajectories spanning tool-use, drawer manipulation, cube rearrangement, and puzzle solving. Each demonstration is generated via scripted policies that guarantee feasibility and success. For the LCBC and LCDP baselines, we further annotate each demonstration with natural-language descriptions of the task. Tese annotations also help evaluate how well language-conditioned models can generalize across start–goal variations.

**Success Condition.** A rollout is considered successful if all objects reach their target configurations within a predefined spatial threshold.

## E.2    TOOL

The Tool scene (Figure 7) requires the robot to manipulate a tool in order to push the cube to the target location. Success is achieved when the cube reaches the designated target area within a fixed distance threshold. Direct manipulation is not possible, so the robot must use the provided tool to accomplish the task.

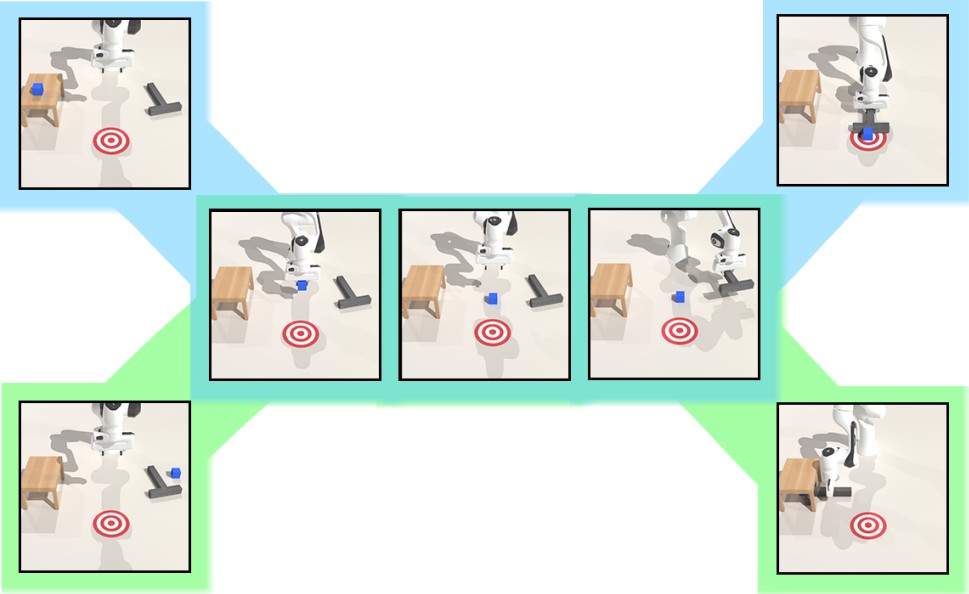

Figure 7: **Visualization of Tool Scene.**

### E.3 DRAWER

The Drawer scene (Figure 8) requires the robot to manipulate drawers into closed states and use the brush to draw on the canvas. Depending on the start state, the brush may be located in different spaces, and the robot must go to the correct space and retrieve it. After grasping the brush, the drawer must often be closed again before drawing, to avoid collision between the arm and the drawer.

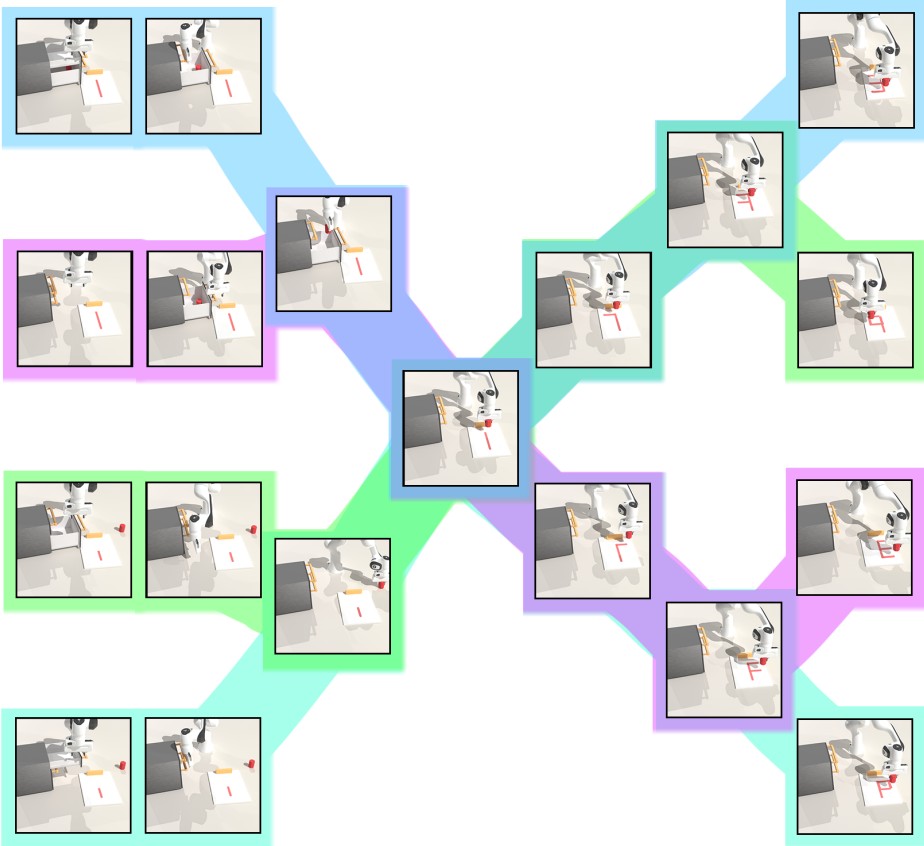

Figure 8: **Visualization of Drawer Scene.**

### E.4 CUBE

The Cube scene (Figure 9) requires the robot to manipulate multiple colored cubes, first arranging them into a prescribed order and then placing each cube into its designated goal region. This task evaluates the planner's ability to identify and distinguish between object colors, maintain the correct ordering, and execute precise placement into multiple targets.

### E.5 PUZZLE

The Puzzle scene (Figure 10) poses the most challenging test of compositional planning. The robot must first arrange multiple colored blocks into a specific intermediate configuration, and then place them into distinct goal slots. This requires not only accurate object manipulation and ordering, but also the ability to chain together multiple sub-tasks that were only observed in isolation during demonstrations.

## F    IMPLEMENTATION DETAILS

**Software.** All experiments are conducted on Ubuntu 20.04.6 with Python 3.10 and PyTorch 2.2.1.

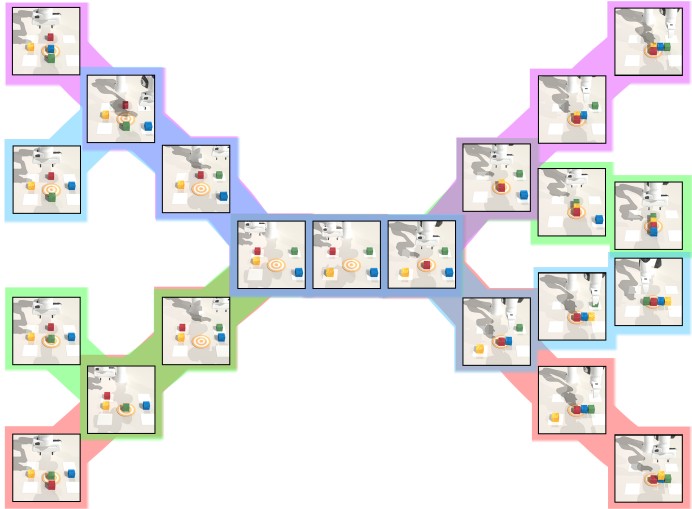

Figure 9: **Visualization of Cube Scene.**

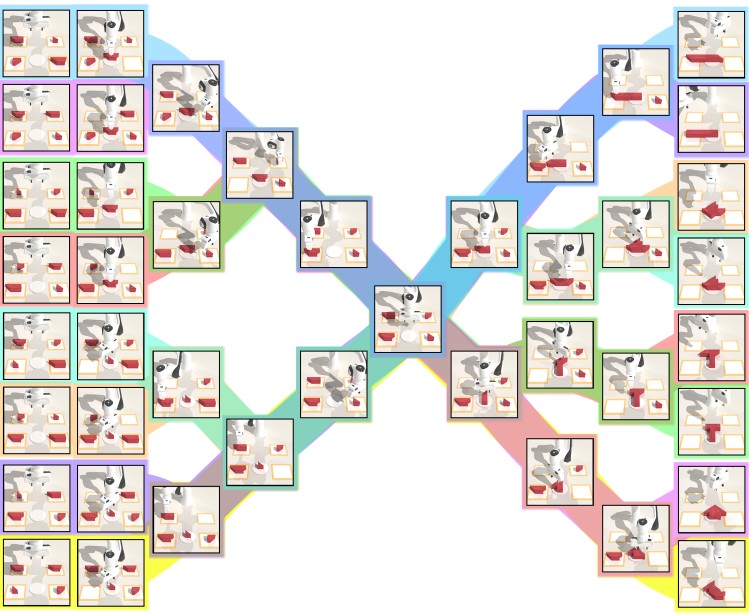

Figure 10: **Visualization of Puzzle Scene.**

**Training.** Models are trained on NVIDIA H200 GPUs.

**Deployment Hardware.** For deployment, we use a single NVIDIA L40S GPU.

**Model Inputs and Outputs.** All observations are first encoded into tokens using the Cosmos tokenizer. We adopt DiT backbones for video generation, using DiT-L or DiT-XL (Peebles & Xie, 2023) depending on the scene. Video generation is performed entirely in the token space. For control, we employ a simple MLP-based inverse dynamics model, which predicts low-level actions conditioned on consecutive frames.

**Hyperparameters.** We report all hyperparameters used during both training and inference for full transparency. For fairness and reproducibility, we do not perform any hyperparameter search or use a learning rate scheduler; all experiments are conducted with fixed values throughout. This ensures that performance gains arise from the method itself rather than extensive hyperparameter tuning. We also keep hyperparameters consistent across different tasks and scenes, unless otherwise specified, to highlight the robustness of our approach.

| Hyperparameter | Value |
|---|---|
| Diffusion Time Step | 500 |
| Batch Size | 512 |
| Optimizer | Adam |
| Learning Rate | $1 \times 10^{-4}$ |
| Iterations | 1M |
| Discount Factor $\gamma$ | 0.6 |
| Sampling Time Steps | 300 |
| Guidance Weight $g_r$ | 0.6 |

Table 4: **Relevant hyperparameters** used in our experiments.

# G  DEPLOYMENT TIME STUDY

## G.1  TIME STUDY & NFES

All results in Table 5 are reported under the same setting of 300 DDIM steps. Compared to DiffCollage, our method incurs higher wall-clock time because it requires test-time backpropagation through the diffusion model in order to enforce consistency via message passing. This extra computation accounts for the increase in sampling time, but is necessary to achieve the significant gains in success rates reported in the main results. Our implementation is fully batched, so all factors require only a single forward pass at each diffusion steps. We follow the standard convention of counting only forward passes of the diffusion model as NFEs. At the same DDIM step count, our method requires twice the NFEs because it performs two forward passes per step. The backward update does not increase NFEs, but it does add a small amount of extra wall-clock time.

| Scene | # Models Composed | Sampling Time (s) ↓ | |
|---|---|---|---|
| | | DiffCollage | Ours |
| Tool-Use | 3 | 7.1 | 17.2 |
| Drawer | 5 | 18.9 | 30.4 |
| Cube | 5 | 21.1 | 31.7 |
| Puzzle | 6 | 30.4 | 61.8 |

Table 5: **Sampling time during deployment.** This is measured as the mean wall-clock time across all samples within a single scene.

# H REAL WORLD QUALITATIVE RESULTS

## H.1 IN DISTRIBUTION EVALUATION

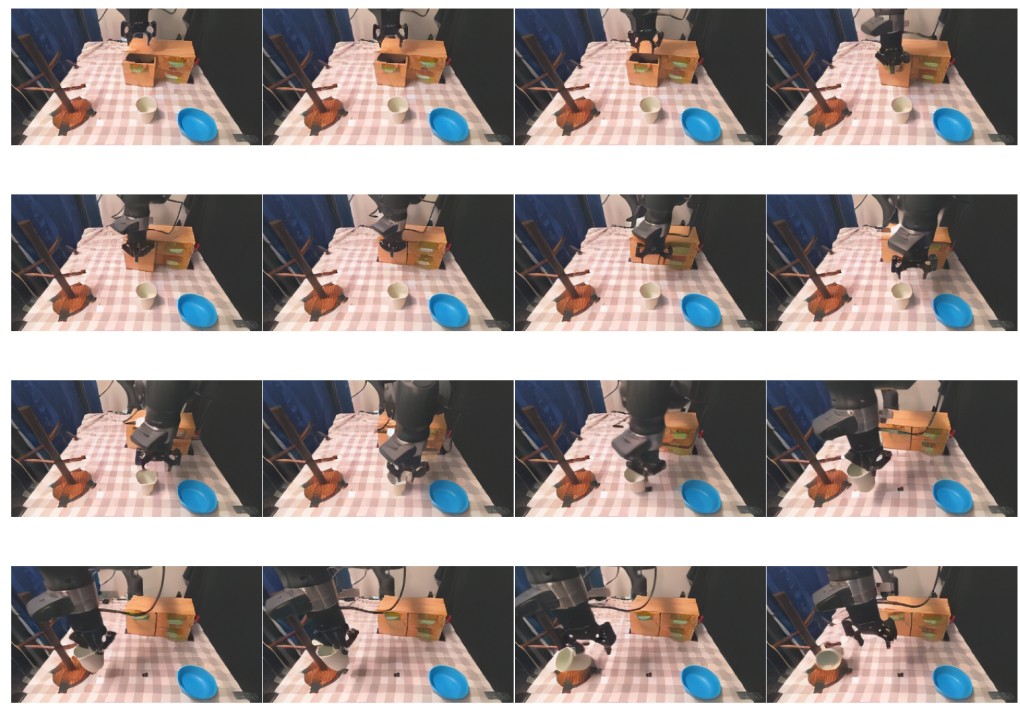

Figure 11: **Task1 Synthetic Plan.**

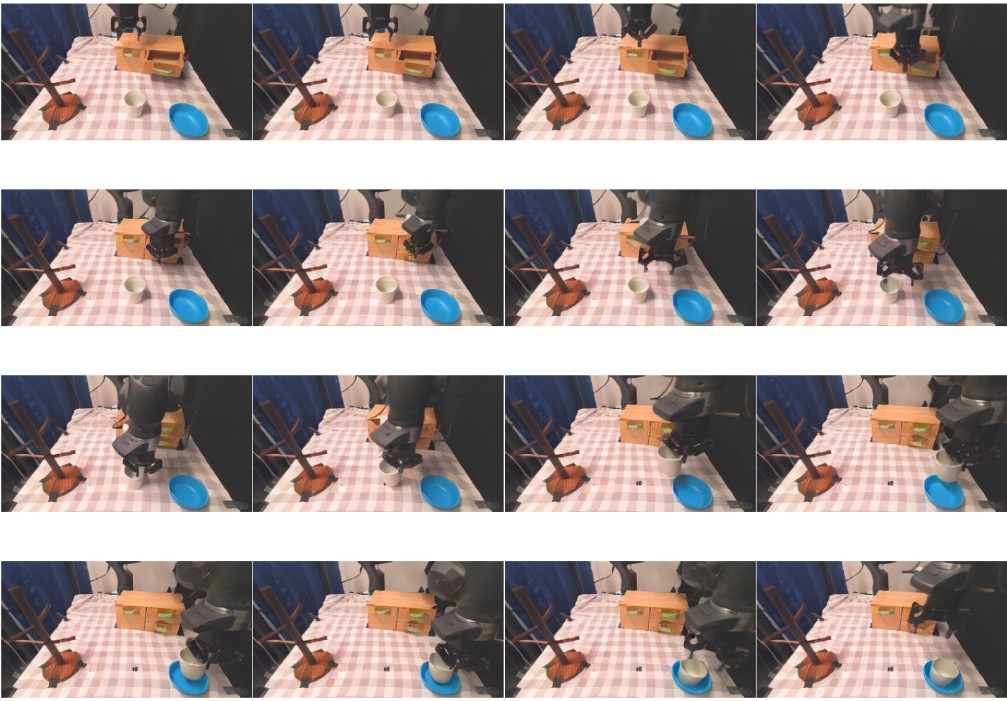

Figure 12: **Task2 Synthetic Plan.**

## H.2    OUT OF DISTRIBUTION EVALUATION

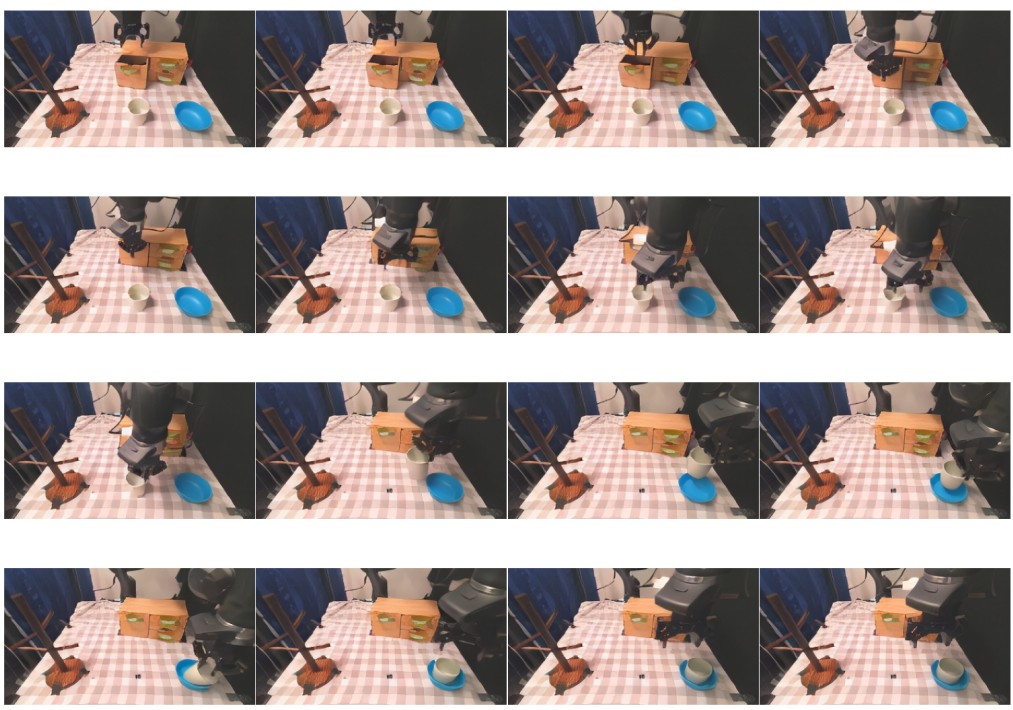

Figure 13: **Task3 Synthetic Plan.**

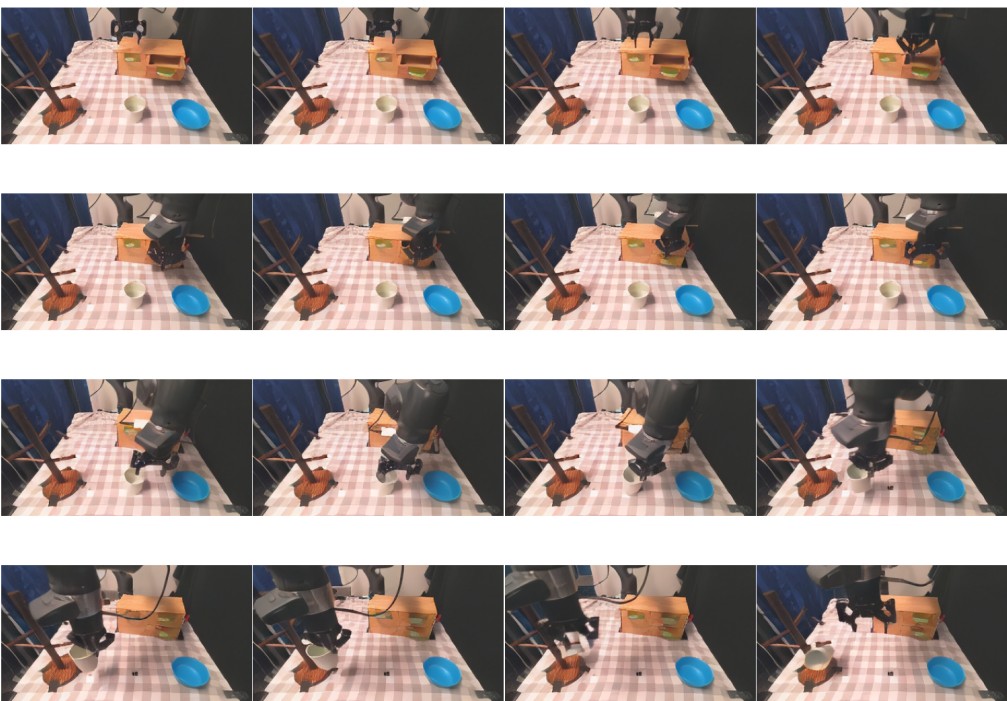

Figure 14: **Task4 Synthetic Plan.**

## I  FAILURE CASE OF SYNCHRONOUS MESSAGE PASSING

In our experiments, removing the discount factor does not cause plans to become globally incoherent, the generated videos remain smooth, and the intermediate states are generally consistent. The main failure mode is more subtle: the generated start and goal frames remain semantically correct but exhibit small spatial misalignments relative to the precise test-time requirement. This misalignment leads to plans that are coherent in motion but anchored to an incorrect spatial configuration.

For example, in our failure cases 1 and 2, the robot performs a reasonable picking motion for the hook, yet the entire sequence is centered around the wrong object location: the gray tool is consistently generated with a small but noticeable orientation offset from its true pose required by start. We hypothesize that without the discount factor, the guidance signals from the start and goal conditions are not sufficiently emphasized, allowing this spatial drift to persist throughout the whole plan.

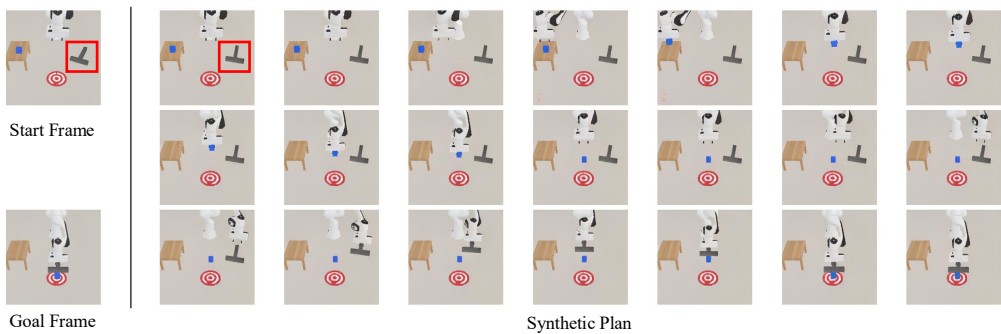

Figure 15: **Failure Mode of Synchronous Message Passing: Case 1.**

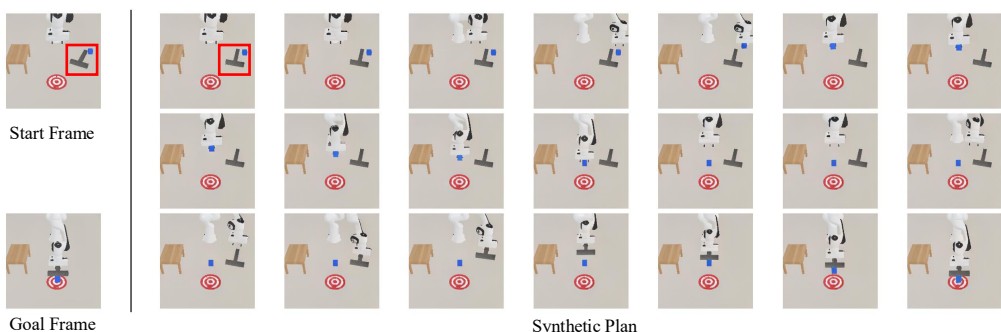

Figure 16: **Failure Mode of Synchronous Message Passing: Case 2.**

