# OpenReview forum: "Compositional Visual Planning via Inference-Time Diffusion Scaling"
_ICLR.cc/2026/Conference — ICLR 2026 Poster_

### Official Review · Reviewer_geNQ · 2025-11-02

**Soundness:** 3
**Presentation:** 3
**Contribution:** 4
**Rating:** 8
**Confidence:** 3

**Summary:**

This paper tackles the challenge of generating long-horizon visual plans using diffusion models trained only on short video clips. The authors introduce CVP, a training-free inference-time framework that composes overlapping short-horizon segments through message passing on denoised (Tweedie) estimates rather than noisy diffusion states. This approach ensures global temporal consistency and smooth transitions across segments without retraining. Experiments on robotic manipulation and visual planning benchmarks show that CVP substantially improves video quality, coherence, and task success rates over prior compositional and policy-based baselines.

**Strengths:**

- Composing diffusion-based video planners in a training-free manner by, shifting consistency enforcement to denoised Tweedie space is novel.
- The proposed factor-graph formulation analysis provide a clear and rigorous justification for the method.
- The approach achieves large performance gains in both visual quality and planning success.
- The framework shows robust out-of-distribution generalization.

**Weaknesses:**

- The experiments, though extensive within robotic planning, are restricted to a single simulation domain (ManiSkill). Testing on real-world data or non-robotic visual domains would strengthen the claim of general applicability. Without at least some transfer or qualitative evidence on real-world data, it remains unclear how robust the approach is.
- The paper could better position its contribution relative to classical planning and hierarchical control methods, making clearer how compositional diffusion planning aligns with them.
- The paper doesn't analyze the computational efficiency or runtime implications of the added message-passing guidance.

**Questions:**

- Since the current formulation assumes a linear chain structure, can the approach be extended to tree-structured or multi-goal planning problems? For example, tasks with subgoals or parallel branches.
- To what extent are the observed gains attributable to the compositional inference versus the base diffusion model quality? Would weaker or stronger base models change the relative benefit?

---

> ### Author Response · Authors · 2025-11-25
> **Rebuttal by Authors**
>
> Thank you for the highly detailed review and constructive feedback about this work.
> > W1. The experiments, though extensive within robotic planning, are restricted to a single simulation domain (ManiSkill). Testing on real-world data or non-robotic visual domains would strengthen the claim of general applicability. Without at least some transfer or qualitative evidence on real-world data, it remains unclear how robust the approach is.
>
> Thank you for the valuable suggestion. Evaluation beyond simulation is important for demonstrating robustness. In the revision, we therefore add a set of real-robot experiments (see Section 5.4 and Appendix H), which serve as a concrete verifier of our method’s capability outside ManiSkill. Quantitative(Section 5.4) and qualitative(Appendix H)resutls shows our compositional planner continues to produce coherent long-horizon behaviors and maintains strong performance in the real world.
>
> |Real Scene| IND:Task1 | IND:Task2 | OOD:Task3 |OOD:Task4|
> |------ |----|----|-----|-----|
> | DiffCollage| 1/10|1/10|0/10|0/10|
> | Ours     | 9/10|7/10|10/10|8/10|
>
> For qualitative evidence for simulation, our anonymous project page (linked in the abstract and here for convenience: https://comp-visual-planning.github.io/) hosts all in-distribution and out-of-distribution synthesized videos and policy rollouts for our method and DiffCollage.
>
> > W2. The paper could better position its contribution relative to classical planning and hierarchical control methods, making clearer how compositional diffusion planning aligns with them.
>
> Thank you for the valuable suggestion. We have incorporated this into the Introduction to better position the contributions of our work in introduction.
> *Classical planning methods, such as Task and Motion Planning (TAMP), decompose tasks into structured subproblems and enforce constraint satisfaction through symbolic operators, while hierarchical control methods first solve for a high-level task plan and then refine it into a low-level motion plan. Compositional diffusion planning follows these core principles but provides a data-driven, probabilistic alternative to hand-engineered classical and hierarchical planners.*
>
> > W3. The paper doesn't analyze the computational efficiency or runtime implications of the added message-passing guidance.
>
> Our initial submission already reports runtime in the appendix F: our sampler is only 1.5×~2x the wall-clock time of a simple DiffCollage-style compositional sampler for 300 DDIM steps. In the revision, we also augment this appendix table with explicit NFE counts.
>
>
> > Q1. Since the current formulation assumes a linear chain structure, can the approach be extended to tree-structured or multi-goal planning problems? For example, tasks with subgoals or parallel branches.
>
> Thank you for the valuable suggestion. For tasks with subgoals,  the agent can reach the goal via either $s \rightarrow a \rightarrow g$ or $s \rightarrow b \rightarrow g$. Because our method is training-free, subgoal requirements can be imposed directly at test time by adding constraints to enforce the chosen path [1].
>
> Extending our method to RRT-style, tree-structured planning is also natural. Our asynchronous objective decomposes into a forward term $L_{\text{fwd}}$ that plans from the start and a backward term $L_{\text{bwd}}$ that plans from the goal. We initialize a forward tree $T_s$ from $s$ and a backward tree $T_g$ from $g$, and then iterate: (1) sample a batch of stitched paths using $L_{\text{fwd}}$ to expand the forward tree, (2) sample a batch of stitched paths using $L_{\text{bwd}}$ to expand the backward tree, and (3) connect $T_s$ and $T_g$ when nodes fall below a certain threshold. We implement this algorithm on OGBench, which achieves 100% success rate on `pointmaze-large`. And we also include this preliminary result in our revised version, and extending it to more complex settings will require additional exploration, which is beyond the scope of this work. More deailts in appendix G.3.
>
> Beyond classical sampling-based planners, our stitching framework naturally extends to Monte Carlo Tree Search as well, and a recent work [2] has already explored in this direction.

---

> ### Author Response · Authors · 2025-11-25
> **Rebuttal by Authors**
>
> > Q2. To what extent are the observed gains attributable to the compositional inference versus the base diffusion model quality? Would weaker or stronger base models change the relative benefit?
>
> Thank you for your valuable suggestion. Our gains are not coming from a stronger base model alone: for all experiments, the baselines and our method share exactly the same diffusion backbone, and we only change the inference procedure.
> Empirically, when we vary the backbone capacity (same architecture, different capacity), we observe that stronger backbones improve the absolute performance of our method, and the baselines remain substantially worse even with the same stronger backbones. This indicates that our contribution is orthogonal to backbone quality and continues to provide benefit even as the base model improves.
> |Tool-IND| DiT-S-P8 | DiT-S-P4 | DiT-B-P8 | DiT-B-P4 |
> |-- |----------|----------|----------|----------|
> |DiffCollage| 0±0 | 0±0 | 0±0 | 1±2 |
> |Ours| 90±5 | 93±2 | 96±2 | 97±3 |
>
> We would like to thank you again for asking important questions about our work. We believe the additional clarifications have strengthened the paper. Please let us know if you have any additional concerns or questions.
>
> [1] Feng, Z., Luan, H., Goyal, P., & Soh, H. (2024). LTLDoG: Satisfying Temporally-Extended Symbolic Constraints for Safe Diffusion-based Planning. IEEE Robotics and Automation Letters, 9.
>
> [2] Yoon, J., Cho, H., & Ahn, S. (2025). Compositional Monte Carlo Tree Diffusion for Extendable Planning. arXiv preprint arXiv:2510.21361.

---

### Official Review · Reviewer_dR6s · 2025-11-04

**Soundness:** 3
**Presentation:** 3
**Contribution:** 2
**Rating:** 2
**Confidence:** 3

**Summary:**

The paper proposes a training-free, inference-time scheme to compose long-horizon visual plans by stitching overlapping short video segments from a pretrained diffusion model. The key idea is to enforce boundary agreement on Tweedie estimates (denoised (x_{0|t})) rather than on noisy states (x_t), implemented via (i) a synchronous Gaussian chain constraint and (ii) an asynchronous, stop-gradient message-passing loss. These constraints are injected into DDIM using diffusion-sphere guidance (DSG). Algorithm 1 describes the sampler. On a simulated benchmark of four manipulation scenes, the method reports higher success rates than a DiffCollage-style baseline and certain policy baselines.

**Strengths:**

- Clear modeling lens: Neat factor-graph formulation and explicit block-tridiagonal precision structure for boundary consistency; derivations are technically correct.

- Practical sampler design: Integrates guidance into DDIM via DSG with synchronized/async constraints; the ablation indicates the async stabilization matters for performance.

- Empirical gains on the paper’s benchmark: Improved temporal coherence metrics and task success relative to selected baselines.

**Weaknesses:**

- Enforcing guidance on (x_{0|t}) is a standard mechanism in training-free guidance; the synchronous GMRF derivation is textbook
- The primary compositional baseline (DiffCollage/GSC) has near-zero success across tasks, making relative gains hard to interpret and raising concerns about implementation/tuning rather than superiority.
- The Noisy-Bethe identity does not establish that Tweedie-space constraints are superior;

**Questions:**

How does joint denoising baseline compare to this method and DiffCollage-style factorization? How much better is it as reference?

---

> ### Author Response · Authors · 2025-11-25
> **Rebuttal by Authors**
>
> We thank the reviewer for the feedback and would like to clarify the positioning of our work within robotics, autonomy, and planning. The motivation of this paper is to identify the limitations of existing compositional planning methods and to propose an orthogonal direction that addresses these issues empirically. Our method empirically offers a practical alternative for long-horizon visual planning.
>
> > W1. Enforcing guidance on ($x_{0|t}$) is a standard mechanism in training-free guidance; the synchronous GMRF derivation is textbook.
>
> We agree that using $x_{0|t}$ for guidance in a **single** diffusion model is standard. However, our setting is different: we care about global consistency across **multiple** overlapping factors in a compositional planning problem, not just guiding one model in isolation. To address this, we cast compositional sampling as global optimization on a chained factor graph and derive a synchronous Gaussian message-passing update that connects classical Gaussian belief propagation with diffusion sampling. However, purely synchronous updates are empirically prone to local minima, so we introduce further a TD-style, one-sided (stop-gradient) update that stabilizes the procedure and significantly accelerates convergence(see Section 4.3 and 5.2 for more details).
>
> > W2. see next response
>
> > W3. The Noisy-Bethe *identity* does not establish that Tweedie-space constraints are superior.
>
> We would like to clarify the role of the Noisy–Bethe *gap* in our argument is to make explicit the structural failure mode of DiffCollage/GSC. We propose an orthogonal design choice to impose constraints directly in Tweedie space, and the superiority of this design is established entensively through empirical results, with additional qualitative evidence (synthetic videos and rollouts) provided on our anonymous project website (linked in the abstract and here for convenience: https://comp-visual-planning.github.io/).

---

> ### Author Response · Authors · 2025-11-25
> **Rebuttal by Authors**
>
> > W2. Concerns about implementation/tuning rather than superiority.
> Q1. How does joint denoising baseline compare to this method and DiffCollage-style factorization? How much better is it as reference?
>
> We understand the concern about baselines, however, the baselines based on official codebase are *carefully tuned and validated*. We'd like to clarify DiffCollage / GSC are *joint denoising* over the full long-horizon trajectory, starting from a single long-horizon Gaussian. Below, we (1) summarize the key differences among all joint-denoising baselines, (2) provide additional empirical evidence verifying the correctness of our baseline implementations, and (3) include an hyperparameter search for existing baselines.
>
> **Joint Denoising Baselines**
> To ensure clarity about how the prior compositional baselines are instantiated in our setting, we briefly summarize the exact formulations of all baselines below.
> * DiffCollage/GSC compose a long-horizon score following Eq. (2) and iteratively denoise the entire long trajectory.
> * DiffCollage+Repainting[1] extends DiffCollage by adding repainting [2] at each denoising step, repeatedly alternating between forward noising and denoising to to increase NFEs.
> * CompDiffuser[3] is a *trained* joint-denoising baseline: it trains the short-horizon model to condition on its preceding and following noisy chunks. At test time, it subdivides the long trajectory into overlapping chunks, and denoises them jointly while conditioning each chunk on its neighbors. CompDiffuser is **SOTA** joint-denoising baseline on OGBench 2D xy point maze navigation; however, its behavior on high-dimensional visual planning had *not* been evaluated prior to our experiments.
>
> We use the official codebases for all baselines:
> * For DiffCollage/GSC, we use the official codebase with recommended hyperparameters and exactly the same pretrained checkpoints as our method, since all methods are training-free in the same setup as ours.
> * For CompDiffuser, we use the official codebase and match the model capacity to ours. And we select the checkpoint with the best validation loss.
>
> **Validation on OGBench Tasks**
> To faithfully verify that our implementations are correct and reasonably tuned, we evaluate on OGBench tasks, where CompDiffuser reports results for both DiffCollage and CompDiffuser. Our reproduced results match the reported performance with only minor statistical variation.
> | Env | DiffCollage/GSC | CompDiffuser | Ours |
> |---------|----------|----------|------------|
> |`pointmaze-medium`     | 100±0  |100±0 |100±0 |
> |`pointmaze-large`      |100±0 |100±0 |100±0 |
> | `pointmaze-giant`       | 32±4  |67±3 |93±4  |
>
> **Baselines with substantially more NFEs**
> On our high-dimensional tool scenes, we perform a hyperparameter search for DiffCollage and its varient over the number of NFEs, allowing many more NFEs(function of evaluations, sampling steps) than our method(300 sampling steps). In these high-dimensional settings, CompDiffuser consistently outperforms DiffCollage and DiffCollage+Repainting.
>
> For DiffCollage,
> |Tool-Scene| NFE 300 | NFE 400 | NFE 500 |
> |--------- |----------|----------|------------|
> |IND| 1±2       | 0±0        |   3±2        |
> |OOD| 0±0       | 2±2       |  1±2     |
>
> For DiffCollage+Repainting with 10 repaint steps,
> |Tool-Scene| NFE 3000 | NFE 4000 | NFE 5000 |
> |--------- |----------|----------|------------|
> |IND| 1±0       | 2±1      |   5±3       |
> |OOD| 1±2       | 3±2       |   2±1       |
>
> For CompDiffuser,
> |Tool-Scene| NFE 300 | NFE 400 | NFE 500 |
> |--------- |----------|----------|------------|
> |IND| 60±2       | 62±3       |   64±3      |
> |OOD| 51±3       | 52±3      |   49±3       |
>
> **CompDiffuser on Visual Planning Tasks** The strongest baseline (CompDiffuser)(table 2 in our revised verison) indeed outperform DiffCollage/GSC, but its performance still degrades as more model composed in more challeging tasks. Our method has substantial gains over the best-performing baseline CompDiffuser.
> |    CompDiffuser      |Tool | Cube | Drawer |  Puzzle |
> |--------- |----------|----------|------------|------------|
> | Num of Model Composed | 3 | 5 | 5 | 6 |
> | Success Rate(IND) |60±2|32±8|20±5|10±3|
> | Success Rate(OOD) |51±3|34±6|18±3|9±3|
>
> |    Ours      |Tool | Cube | Drawer |  Puzzle |
> |--------- |----------|----------|------------|------------|
> | Num of Model Composed | 3 | 5 | 5 | 6 |
> | Success Rate(IND) |97±3|64±10|53±5|50±11|
> | Success Rate(OOD) |96±2|65±9|52±6|50±13|
>
> We believe the additional results and clarifications have strengthened the paper and better addressed the concern. Please let us know if you have any further questions.
>
> [1] Anonymous. (2025). Compositional Diffusion with Guided Search for Long-Horizon Planning. under review.
>
> [2] RePaint: Inpainting using Denoising Diffusion Probabilistic Models. arXiv preprint arXiv:2201.09865.
>
> [3] Generative Trajectory Stitching through Diffusion Composition. arXiv preprint arXiv:2503.05153.

---

### Official Review · Reviewer_bi53 · 2025-11-04

**Soundness:** 4
**Presentation:** 3
**Contribution:** 3
**Rating:** 8
**Confidence:** 4

**Summary:**

This paper introduces a novel, training-free method for compositional planning leveraging diffusion models. The authors begin by theoretically interrogating the inaccuracies of prior compositional generation methods, such as DiffCollage, which rely on a Bethe approximation in the noised state. This inaccuracy is formalized in the paper as the "Noisy-Bethe Gap Theorem" (Theorem 1). To address this limitation, the paper proposes a novel inference-time guidance technique involving synchronous and asynchronous message passing. This mechanism is designed to enforce boundary agreement directly on the estimated Tweedie estimates (the predicted $x_0$​) rather than on the noisy intermediate states.

The proposed method is validated on a suite of robotic manipulation tasks. This is achieved by first generating long-horizon visual plans (scenes) using the diffusion model, and subsequently inferring executable actions via a separately trained inverse dynamics model. Furthermore, the authors conduct a comparative analysis against DiffCollage, evaluating not only task success but also the quality of the generated visual plans using metrics such as motion smoothness, background consistency, and aesthetic quality.

The results demonstrate that the proposed framework significantly outperforms the baseline (DiffCollage) across both task success rates and scene generation quality. Specifically, while DiffCollage fails on nearly all manipulation tasks, the proposed method achieves success rates ranging from 50-100%. Moreover, it demonstrates roughly a two-fold improvement in key video quality metrics like motion smoothness and background consistency.

The authors include ablation studies on the components of their novel composition rule (i.e., synchronous vs. asynchronous message passing) and the number of denoising steps. A practical comparison of sampling time against the baseline is provided in Appendix F, offering readers a concrete understanding of the computational trade-offs. The appendices also furnish the theoretical derivations for the inaccuracy of the noisy Bethe approximation (Theorem 1) and the formulation of synchronous message passing (Theorem 2). Finally, the authors provide comprehensive details on the experimental environments, tasks, and model hyperparameters used for validation.

**Strengths:**

- **Clear Articulation of Problem and Strong Theoretical Grounding:** The paper's primary strength lies in its clear theoretical articulation of a critical problem in compositional planning. The authors compellingly argue that prior methods (e.g., DiffCollage) are flawed due to their reliance on a Bethe approximation in the noisy state. This critique is not just qualitative; it is substantially supported by the "Noisy-Bethe Gap Theorem" (Theorem 1, detailed in Appendix A), which provides a strong formal foundation for the work.

- **Significant Empirical Gains and Thorough Analysis:** The proposed method demonstrates a striking and significant performance gap compared to the baseline. While DiffCollage fails on nearly all robotic manipulation tasks, the proposed method achieves high success rates (50-97%). This empirical success is well-supported by a thorough ablation study (Figures 4 and 5) that effectively isolates the contributions of the synchronous and asynchronous message passing components. The analysis also practically demonstrates how performance scales positively with increased denoising steps (up to 300).

- **Completeness and High Potential for Reproducibility:** The authors are commended for the paper's completeness. Beyond the main theorem, they provide a detailed theoretical derivation for the synchronous message passing mechanism (Appendix B) . This theoretical rigor, combined with comprehensive details on the simulation environments, tasks (Appendix D) , and hyperparameters (Appendix E) , ensures a high potential for reproducibility.

**Weaknesses:**

- **Insufficient Discussion on the Nature of the Contribution:** The paper attributes its success to composing in the Tweedie ($x_0$​ estimate) space. However, the powerful composing rules ($L_\text{sync}$​ / $L_\text{async}$​) are themselves very strong, explicit guidance heuristics. A key insight may be that the Tweedie estimation space is the first domain that enables such strong, explicit guidance to be applied stably (whereas it would fail in the noisy state). The authors are encouraged to reframe their contribution to reflect this: composing on clean estimates provides not just better Bethe approximation but also a crucial platform for powerful heuristics, and their proposed message passing is one such successful instance.

- **Potential for Generating Unrealistic Plans:** One of unaddressed limitations is the potential for the guidance to create unrealistic plans. The method guides sampling to enforce only boundary smoothness between short plans. It is plausible that this guidance could force the generation of a physically impossible trajectory within a short plan if the boundaries are in difficult-to-connect states (e.g., on opposite sides of a wall). The system might prioritize a smooth interpolation (e.g., passing through the obstacle) over maintaining physical plausibility.

- **Lack of Adaptivity in Plan Length:** As the authors acknowledge in their limitations, the method lacks adaptivity in plan length, requiring the number of factors ($n$) to be manually specified. This prevents the planner from dynamically adjusting to task complexity, potentially over-planning for simple tasks or under-planning for difficult ones.

- **(Minor) Presentation and Clarity Issues:**

    - **Equation 5:** the second term's numerator appears to have a typo. It is written as $x_t​−\sqrt{\bar{\alpha}_t}x_0$, but based on the formulation in Equation 3, it should likely be ​$x_t​−\sqrt{\bar{\alpha}_t}x_{0|t}$.

    - **Figure 2 Caption:** The terms “DiffCollage” and "GSC" are used without any prior definition. Furthermore, "Diffcollage" should be capitalized consistently as "DiffCollage" .

    - **Qualitative Comparison:** The paper would greatly benefit from a direct qualitative comparison (e.g., side-by-side frames) showing the failure modes of DiffCollage (e.g., blurriness, incoherence) against the successful generations of the proposed method.

    - **Table 2:** The spacing between the caption and the table is too tight, harming readability.

- **Compliance with ICLR Policy:** The manuscript does not appear to include the mandatory statement on the use of Large Language Models (LLMs) in its preparation, as required by the ICLR 2026 Author Guide.

**Questions:**

- **Rationale for Discount Factor (Eq. 11):** Regarding the asynchronous loss, why is a discount factor γ (set to 0.6) used to down-weight connections far from the start and goal? Intuitively, temporal smoothness and boundary agreement would seem equally important across the entire trajectory, not just near the endpoints. Could the authors elaborate on this design choice?

- **Performance Saturation (Fig. 5):** In Figure 5, the performance on the Drawer scene is still clearly improving at 300 sampling steps. Have the authors explored running this experiment with more steps until performance saturates? It would be valuable to know if it plateaus, or if performance eventually degrades.

- **Sensitivity to Loss Weighting (Fig. 4):** The ablation in Figure 4 compares "Sync Only", "Async Only", and their combination. This reviewer is curious about the sensitivity to the  weighting between these two losses. What would a sweep of $w \cdot L_\text{sync}​+(1−w) \cdot L_\text{async}$​ reveal? This would provide insight into the robustness of the method and the complementary nature of the two losses.

---

> ### Author Response · Authors · 2025-11-25
> **Rebuttal by Authors**
>
> Thank you for the highly detailed review and constructive feedback about this work.
> > W1. Insufficient Discussion on the Nature of the Contribution: The paper attributes its success to composing in the Tweedie space. However, the powerful composing rules  are themselves very strong, explicit guidance heuristics. A key insight may be that the Tweedie estimation space is the first domain that enables such strong, explicit guidance to be applied stably (whereas it would fail in the noisy state). The authors are encouraged to reframe their contribution to reflect this: composing on clean estimates provides not just better Bethe approximation but also a crucial platform for powerful heuristics, and their proposed message passing is one such successful instance.
>
> Thank you for the valuable suggestion. We have incorporated this into the Introduction to better position the contributions of our work.
>
> > W2. Potential for Generating Unrealistic Plans: One of unaddressed limitations is the potential for the guidance to create unrealistic plans. The method guides sampling to enforce only boundary smoothness between short plans. It is plausible that this guidance could force the generation of a physically impossible trajectory within a short plan if the boundaries are in difficult-to-connect states (e.g., on opposite sides of a wall). The system might prioritize a smooth interpolation (e.g., passing through the obstacle) over maintaining physical plausibility.
>
> Thank you for the insightful question. We would like to clarify that our method does not forcibly stitch short plans together. The boundary consistency is imposed through a *soft* guidance penalty, not a *hard* constraint. Our DSG-based guidance preserves the underlying data manifold, meaning that if the demonstrations never exhibit behaviors such as “passing through a wall,” the guided sampler has very little chance of generating such unrealistic trajectories.
>
> Moreover, because our framework is fully training-free, we can easily incorporate additional physical constraints when available: for example, adding an explicit collision penalty $L_{collision}$ using a known environment map. This further prevents generating physically impossible plans. More details in appendix G.2.
> | Env | Ours | Ours + $L_{collision}$ |
> |---------|----------|----------|
> |`pointmaze-medium` | 100±0     | 100±0       |
> |`pointmaze-large`  | 100±0     | 100±0       |
> | `pointmaze-giant` | 93±4      | 97±3       |
> > W3. Lack of Adaptivity in Plan Length: As the authors acknowledge in their limitations, the method lacks adaptivity in plan length, requiring the number of factors to be manually specified. This prevents the planner from dynamically adjusting to task complexity, potentially over-planning for simple tasks or under-planning for difficult ones.
>
> We acknowledge this is a limitation and have already included this in the limitation section. We note that this limitation is shared by existing compositional diffusion methods (e.g., DiffCollage, GSC, CompDiffuser). However, we can sidestep this issue by integrating our approach with classical planning.
>
> Extending our method to RRT-style, tree-structured planning is natural(appendix G.3). Our asynchronous objective decomposes into a forward term $L_{\text{fwd}}$ that plans from the start and a backward term $L_{\text{bwd}}$ that plans from the goal. We initialize a forward tree $T_s$ from $s$ and a backward tree $T_g$ from $g$, and then iterate: (1) sample a batch of stitched paths using $L_{\text{fwd}}$ to expand the forward tree, (2) sample a batch of stitched paths using $L_{\text{bwd}}$ to expand the backward tree, and (3) connect $T_s$ and $T_g$ when nodes fall below a certain threshold. We implement this algorithm on OGBench, which achieves 100% success rate on `pointmaze-large`. And we also include this preliminary result in our revised version, and extending it to more complex settings will require additional exploration, which is beyond the scope of this work.

---

> ### Author Response · Authors · 2025-11-25
> **Rebuttal by Authors**
>
> > W4.1. Equation 5: the second term's numerator appears to have a typo. It is written as
> , but based on the formulation in Equation 3, it should likely be ​$x_t​−\sqrt{\bar{\alpha}_t}x_{0|t}$.
> W4.2. Figure 2 Caption: The terms “DiffCollage” and "GSC" are used without any prior definition. Furthermore, "Diffcollage" should be capitalized consistently as "DiffCollage" .
> W4.3 Table 2: The spacing between the caption and the table is too tight, harming readability.
> W5: LLM usage
>
> Thank you for the valuable suggestion. We have updated the revised version accordingly.
>
> > Q1. Rationale for Discount Factor (Eq. 11): Regarding the asynchronous loss, why is a discount factor γ (set to 0.6) used to down-weight connections far from the start and goal? Intuitively, temporal smoothness and boundary agreement would seem equally important across the entire trajectory, not just near the endpoints. Could the authors elaborate on this design choice?
>
> Thank you for the insightful question. In our design, temporal smoothness and boundary agreement are treated uniformly in $L_{sync}$. However, optimizing this global objective directly is prone to local minima. The discount factor $\gamma$ in $L_{async}$ provides a natural way to stabilize the updates: near the start, the update is biased toward information from the start state; near the goal, it is biased toward the goal state. This directional weighting improves optimization stability while still preserving global consistency.
>
> > Q2. Performance Saturation (Fig. 5): In Figure 5, the performance on the Drawer scene is still clearly improving at 300 sampling steps. Have the authors explored running this experiment with more steps until performance saturates? It would be valuable to know if it plateaus, or if performance eventually degrades.
>
> Thank you for the valuable suggestion. As we increase the sampling steps, we observe that the performance eventually plateaus.
> |Steps| 50 | 100 | 200 | 300 | 400 | 500 |
> |-----|----|-----|-----|-----|-----|-----|
> |IND  | 35 | 40  | 45  | 53  | 55  | 54  |
> |OOD  | 20 | 25  | 45  | 52  | 55  | 56  |
>
> > Q3. Sensitivity to Loss Weighting (Fig. 4): The ablation in Figure 4 compares "Sync Only", "Async Only", and their combination. T.
>
> Thank you for the valuable suggestion. We observe that the intermediate weighting region yields the best performance, indicating the effectiveness of our joint message passing algorithm.
>
> | w   | 1.0| 0.8 |  0.6 | 0.5 | 0.4 | 0.2 |0.0|
> |-----|----|-----|----- |-----|-----|-----|---|
> |IND  | 10 | 39  | 66 | 64  | 68  | 60  | 45 |
> |OOD  | 8 | 36  | 60  | 65 | 66  | 61  | 41 |

---

> > ### Comment · Reviewer_bi53 · 2025-11-25
> >
> > Thank you for the detailed rebuttal addressing my concerns and questions. In particular, the ablation results on loss weighting were new and insightful. I still have a remaining question regarding the discount factor $\gamma$. I agree that enforcing boundary agreement uniformly may lead to local minima. Have the authors empirically observed this failure mode in practice?

---

> ### Author Response · Authors · 2025-11-25
>
> Thank you for the thoughtful question. In our experiments, removing the discount factor does not cause plans to become globally incoherent. The generated videos remain smooth and the intermediate states are generally consistent. The main failure mode we observe is more subtle: the generated start and goal frames remain semantically correct but exhibit slight spatial misalignment relative to the precise test-time requirement, causing the entire plan to be guided under a small but consistent shift. For example, the generated video shows a smooth and plausible cube-picking motion, but the object and gripper are slightly shifted relative to the required scene configuration, so the prediction no longer satisfies the exact initial/goal set used for evaluation.
>
> This issue becomes critical in manipulation tasks, where even small spatial deviations in the end effector or object pose can turn a reasonable plan into a failure. Introducing the discount factor helps mitigate this by placing stronger emphasis on *spatial information* from the start and goal, which empirically reduces these fine grained spatial drifts.
>
> We also include the failure visualization in appendix I. The robot performs a reasonable picking motion for the hook,
> yet the entire sequence is centered around the wrong object location: the gray tool is consistently
> generated with a small but noticeable orientation offset from its true pose required by start.
>
> We hope this visualization gives you a clear sense of the problem!

---

> > ### Comment · Reviewer_bi53 · 2025-11-28
> >
> > Thank you for the detailed explanations and the additional visualizations. The authors’ responses satisfactorily address my concerns and questions. I will maintain my current score.

---

### Official Review · Reviewer_PERF · 2025-11-05

**Soundness:** 3
**Presentation:** 3
**Contribution:** 3
**Rating:** 6
**Confidence:** 3

**Summary:**

This paper proposes an inference-time compositional technique to extend existing short-horizon diffusion models to long-horizon robot planning without retraining. To solve the instability issue of prior work (e.g. DiffCollage/GSC) that composes segments in the noisy space ($x_t$)—a problem the paper theoretically identifies as the "Noisy-Bethe Gap" —this method enforces boundary consistency directly on the clean data estimates ($x_{0∣t}$). It achieves this by defining Synchronous ($L_{sync}$) and Asynchronous ($L_{async}$) message passing losses on $x_{0∣t}$ and guides the sampling process using Diffusion-Sphere Guidance (DSG).

**Strengths:**

1. While using $x_0$ prediction (Tweedie estimates) for guidance is not new, its novel application to compositional generation is a impressive contribution, using it to enforce boundary consistency between factors.
2. The paper provides a strong theoretical justification for why prior methods fail. By identifying and proving the "Noisy-Bethe Gap" (Theorem 1) , it formally explains that simple averaging in the noisy $x_t$​ space is fundamentally flawed.
3. It addresses a critical and timely problem: how to achieve inference-time scaling by stitching pre-trained, short-horizon models to generate long-horizon content. This training-free, "plug-and-play" approach is highly practical and flexible, extending the capabilities of existing models without costly retraining.

**Weaknesses:**

1. The justification for using Diffusion-Sphere Guidance (DSG) specifically is not entirely clear. The paper would be significantly strengthened by an ablation study that compares DSG to standard, simpler guidance mechanisms (e.g., conventional gradient-based guidance or other proposed guidance methods). This would help demonstrate that the proposed method is general enough to improve performance even when paired with various other guidance techniques, not just DSG.
2. The paper's claims regarding compositional performance would be significantly strengthened by a broader comparison against other diffusion-based stitching and planning methods. The current experiments are heavily focused on DiffCollage/GSC. Including comparisons to other relevant works, such as CompDiffuser [1], would provide a more comprehensive validation.


[1] Luo, Y., Mishra, U. A., Du, Y., & Xu, D. (2025). Generative trajectory stitching through diffusion composition. arXiv preprint arXiv:2503.05153.

**Questions:**

1. The justification for using Diffusion-Sphere Guidance (DSG) specifically is not entirely clear. Is there any reason that this paper uses the DSG instead of other guidance methods?

2. I am curious about the qualitative adherence of the generated plans to the method's design. When the authors inspected the planning results, did they observe that the plans actually follow the underlying compositional structure (i.e., the chain of factors) described in the appendix? To put this another way: if a setting existed with a metric that specifically measures this 'compositionality,' would the authors hypothesize that proposed method would achieve the best score compared to the baselines?

3. The proposed method introduces significant inference overhead by requiring back-propagation at every DDIM step. The paper lacks a comparison under a fixed same computational budget. For example, does the 300-step proposed method (with costly guidance) outperform a baseline like DiffCollage allowed to run for many more steps (e.g., 1000 or 3000 steps) such that the total wall-clock time/Number of Function Evalutations (NFEs) are equivalent?

---

> ### Author Response · Authors · 2025-11-25
> **Rebuttal by Authors**
>
> Thank you for the highly detailed review and constructive feedback about this work.
>
> > W1. The justification for using Diffusion-Sphere Guidance (DSG) specifically is not entirely clear. The paper would be significantly strengthened by an ablation study that compares DSG to standard, simpler guidance mechanisms (e.g., conventional gradient-based guidance or other proposed guidance methods). This would help demonstrate that the proposed method is general enough to improve performance even when paired with various other guidance techniques, not just DSG.
> Q1. The justification for using Diffusion-Sphere Guidance (DSG) specifically is not entirely clear. Is there any reason that this paper uses the DSG instead of other guidance methods?
>
> Thank you for the valuable suggestion. We choose DSG because conventional gradient-based guidance (DPS) relies on point score estimation, which can deviate from the true score as shown in [1, 2], whereas DSG constrains each guidance step to an intermediate data manifold via an optimization, mitigating this error. For completeness, we also include LGD [3], which replaces DPS’s point-score estimate with a Monte Carlo estimator. Empirically, our method is agnostic to the specific guidance operator: DPS, LGD, and DSG all perform reasonably well, but DSG consistently yields the most stable results, so we adopt it as the default in our main experiments.
> |Tool-Scene| DPS | LGD | DSG |
> |--------- |----------|----------|------------|
> |IND|90±2|95±3| 97±3 |
> |OOD|88±3|92±4|96±3  |
>
>
> > W2. The paper's claims regarding compositional performance would be significantly strengthened by a broader comparison against other diffusion-based stitching and planning methods. The current experiments are heavily focused on DiffCollage/GSC. Including comparisons to other relevant works, such as CompDiffuser, would provide a more comprehensive validation.
>
> Thank you for the valuable suggestion. We use the official codebase for CompDiffuser and match the model capacity to ours. And we select the checkpoint with the best validation loss. CompDiffuser outperform DiffCollage/GSC, but its performance still degrades  as more model composed in more challeging tasks. (table 2 in our revised verison)
>
>
> |    CompDiffuser      |Tool | Cube | Drawer |  Puzzle |
> |--------- |----------|----------|------------|------------|
> | Num of Model Composed | 3 | 5 | 5 | 6 |
> | Success Rate(IND) |60±2|32±8|20±5|10±3|
> | Success Rate(OOD) |51±3|34±6|18±3|9±3|
>
> |    Ours      |Tool | Cube | Drawer |  Puzzle |
> |--------- |----------|----------|------------|------------|
> | Num of Model Composed | 3 | 5 | 5 | 6 |
> | Success Rate(IND) |97±3|64±10|53±5|50±11|
> | Success Rate(OOD) |96±2|65±9|52±6|50±13|
>
>
> > Q2. I am curious about the qualitative adherence of the generated plans to the method's design. When the authors inspected the planning results, did they observe that the plans actually follow the underlying compositional structure (i.e., the chain of factors) described in the appendix? To put this another way: if a setting existed with a metric that specifically measures this 'compositionality,' would the authors hypothesize that proposed method would achieve the best score compared to the baselines?
>
> Yes, we consistently observe that our method exhibits best compositionality. The environments are designed to test this property: in the tool-use and drawer scenes, the agent must first fetch the tool or brush before it can push objects or draw on the canvas; in the cube and puzzle-assembly scenes, success requires completing each intermediate step in the correct order. Our method reliably produces plans that are physically realistic and compositionally consistent. The failure cases typically arise from small action-prediction errors accumulating by inverse dynammics model during execution, since we intentionally disable replanning in order to evaluate the benefit coming from the algorithm itself, rather than from repeated replanning.
>
> For qualitative evidence, our anonymous project page (linked in the abstract and here for convenience: https://comp-visual-planning.github.io/) hosts all in-distribution and out-of-distribution synthesized videos and policy rollouts for our method and DiffCollage.
>
> [1] Diffusion Posterior Sampling for General Noisy Inverse Problems
>
> [2] Guidance with Spherical Gaussian Constraint for Conditional Diffusion
>
> [3] Loss-Guided Diffusion Models for Plug-and-Play Controllable Generation

---

> ### Author Response · Authors · 2025-11-25
> **Rebuttal by Authors**
>
> > Q3. The proposed method introduces significant inference overhead by requiring back-propagation at every DDIM step. The paper lacks a comparison under a fixed same computational budget. For example, does the 300-step proposed method (with costly guidance) outperform a baseline like DiffCollage allowed to run for many more steps (e.g., 1000 or 3000 steps) such that the total wall-clock time/Number of Function Evalutations (NFEs) are equivalent?
>
> Since the diffusion model was trained with 500 DDPM steps, we compare baselines at NFEs from 300 up to 500. DiffCollage+Repainting [1] extends DiffCollage by adding repainting [2] at each denoising step, repeatedly alternating forward noising and denoising to increase NFEs. We report DiffCollage+Repainting results with much larger budgets, from 3000 to 5000 NFEs, using 10 repaint iterations per denoising step. Even with this significant increase in NFEs, the noisy-space factorization in DiffCollage/GSC and DiffCollage+Repainting still breaks down no matter how many NFEs it is: success rates remain low and the generated videos are blurry and temporally inconsistent. In contrast, with 300 DDIM steps our method only costs about 1.5×~2x the wall-clock time of DiffCollage at 300 steps, while achieving substantially better performance.
>
> For DiffCollage,
> |Tool-Scene| NFE 300 | NFE 400 | NFE 500 |
> |--------- |----------|----------|------------|
> |IND| 1±2       | 0±0        |   3±2        |
> |OOD| 0±0       | 2±2       |  1±2     |
>
> For DiffCollage with 10 repaint steps,
> |Tool-Scene| NFE 3000 | NFE 4000 | NFE 5000 |
> |--------- |----------|----------|------------|
> |IND| 1±0       | 2±1      |   5±3       |
> |OOD| 1±2       | 3±2       |   2±1       |
>
> We would like to thank you again for raising important questions about this work, and we believe the additional clarifications and results have strengthened the paper. Please let us know if you have any additional concerns or questions.
>
> [1] Anonymous.  Compositional Diffusion with Guided Search for Long-Horizon Planning.  under review.
>
> [2] Lugmayr, A., Danelljan, M., Romero, A., Yu, F., Timofte, R., & Van Gool, L. (2022). RePaint: Inpainting using Denoising Diffusion Probabilistic Models. arXiv preprint arXiv:2201.09865.

---

### Official Review · Reviewer_zF3D · 2025-11-10

**Soundness:** 4
**Presentation:** 3
**Contribution:** 3
**Rating:** 8
**Confidence:** 4

**Summary:**

This paper builds on DiffCollage, but extends it from image composition to visual planning: instead of stitching small image patches into a larger picture, it stitches short video clips into coherent, long-horizon visual plans. The motivation is to overcome diffusion planners’ inherent limitation: while they excel at short-horizon trajectory generation, scaling them to long-horizon tasks leads to instability and broken temporal consistency when naively composing multiple short plans.

Problem Setting
- Long-horizon robot planning using diffusion models suffers from compounding noise and the breakdown of factorization assumptions when combining multiple short rollouts.
- Prior compositional methods (e.g., DiffCollage, Generative Skill Chaining) operate in noisy latent space, leading to discontinuities at segment boundaries and failure to generalize to unseen start–goal pairs.

Method: Compositional Visual Planning (CVP in short)
- CVP reframes long-horizon planning as inference over a chain-structured factor graph built from overlapping short-horizon video segments.
- Key insight: enforce boundary consistency on Tweedie estimates (denoised predictions) rather than on noisy diffusion states.
- Two complementary message-passing schemes ensure global coherence: (a) Synchronous message passing: treats all factors jointly as a Gaussian linear system for order-invariant, parallel consistency enforcement. (b) Asynchronous message passing: uses one-sided, stop-gradient updates (TD-style) for faster and more stable convergence.
- The two are combined via Diffusion-Sphere Guidance (DSG) to interpolate between unconditional sampling and guided refinement — preserving both stability and diversity.
- Importantly, the method is training-free: it reuses a pretrained short-horizon diffusion backbone without fine-tuning.

The authors evaluate the proposed approach on 100 robotic manipulation tasks across four scenes (Tool-Use, Drawer, Cube, Puzzle) built from ManiSkill.
- In terms of visual quality, CVP achieves smoother motion, consistent backgrounds and better image quality quantified by VBench++ metrics.
- When tested for task success rate (when combined with a inverse dynamics model), CVP achieves major improvements in out-of-distribution generalization, and stronger in-domain performance than existing diffusion policy baselines.

**Strengths:**

- The paper presents a clear conceptual insight: enforcing boundary agreement on denoised Tweedie estimates rather than noisy diffusion states. This shift is both theoretically justified and empirically validated.
- The toy example in Figure 2 effectively illustrates the key failure mode of noisy-state composition (boundary drift) and how the proposed approach closes those gaps. It’s an unusually clear and intuitive visualization of the underlying problem.
- The proposed framework is training-free, operating entirely at inference time, which makes it easy to apply on top of existing short-horizon diffusion backbones.
- The empirical results demonstrate strong improvements in temporal coherence and OOD generalization across a large set of manipulation tasks, validating both stability and scalability.
- The method has solid theoretical grounding, bridging diffusion models with message-passing in factor graphs, which adds conceptual depth beyond heuristic composition.

**Weaknesses:**

- While OOD performance is impressive, the method still relies on the base video diffusion model having seen the intermediate motion fragments during training. The generalization is compositional rather than truly extrapolative, which limits deployment in fully novel environments.
- The method requires a goal image as conditioning, restricting applicability to scenarios where both start and goal visual states are available. Many planning settings might not have explicit goal frames.
- The approach introduces extra inference-time computation due to gradient-based guidance and message-passing updates. The paper lacks runtime comparisons with simpler compositional samplers.
- The paper covers the preliminaries of factor-graph very quickly in section 3.1. Also. the way equation 1 puts the denominator as negative exponent makes it way less intuitive. This limits the accessibility for readers not already familiar with DiffCollage.

**Questions:**

- Language conditioning: Could this approach generalize to language-conditioned visual planning? In other words, can the boundary or goal condition g be replaced by a text or semantic embedding that specifies the desired outcome, rather than an explicit goal image?
- Boundary artifacts in DiffCollage: Figure 2 shows DiffCollage leaving visible boundary gaps, but the original DiffCollage image-generation results generally do not show such artifacts. Could this discrepancy arise because the noisy factorization assumption breaks more severely in video or temporal domains than in static image composition? A short clarification would help reconcile this difference.

---

> ### Author Response · Authors · 2025-11-25
> **Rebuttal by Authors**
>
> Thank you for the highly detailed review and constructive feedback about this work.
> > W1. While OOD performance is impressive, the method still relies on the base video diffusion model having seen the intermediate motion fragments during training. The generalization is compositional rather than truly extrapolative, which limits deployment in fully novel environments.
>
> Thank you for the valuable comment. This limitation is shared by existing video diffusion planning framework [1,2]. Our work takes a step forward to extent them to compostional generation. Moving forward, we hypothesize that stronger base video models or richer visual representations can enable truly extrapolative generalization in fully novel environments, which would be complementary to our framework.
>
> > W2. The method requires a goal image as conditioning, restricting applicability to scenarios where both start and goal visual states are available. Many planning settings might not have explicit goal frames.
>
> Thank you for the valuable comment. When goal visual states are not available, we can synthesize goal images using an image inpainting model[3,4] or a VLM[5]. More broadly, because our method is training-free given modular diffusion models, it also extends naturally to language-conditioned goals without any additional training, We report results in language-conditional planning in response to Q1.
>
> > W3. The approach introduces extra inference-time computation due to gradient-based guidance and message-passing updates. The paper lacks runtime comparisons with simpler compositional samplers.
>
> Our initial submission already reports runtime in the appendix F: our sampler(batched implementation same as baselines) is only 1.5×~2x the wall-clock time of a simple DiffCollage-style compositional sampler for 300 DDIM steps. In the revision, we also augment this appendix table with explicit NFE counts.
>
> > W4. The paper covers the preliminaries of factor-graph very quickly in section 3.1. Also. the way equation 1 puts the denominator as negative exponent makes it way less intuitive. This limits the accessibility for readers not already familiar with DiffCollage.
>
> Thank you for the valuable comment. In the revision, we expand Sec. 3.1 with a concrete linear chain example, so readers need no prior exposure to DiffCollage. We also rewrite Eq. (1) in a intuitive way for readibility.
>
> > Q1. Language conditioning: Could this approach generalize to language-conditioned visual planning? In other words, can the boundary or goal condition g be replaced by a text or semantic embedding that specifies the desired outcome, rather than an explicit goal image?
>
> Thank you for the suggestion. Conceptually, the boundary condition is an important way to extend the planning horizon, and in our training-free setting the goal image can be directly replaced by a text description. Concretely, we define a CLIP-style guidance loss $L_{clip}$ between the generated goal frame at each denoising step and the text prompt, and drop the original goal-image term in $L_{sync}$ and $L_{async}$. In a preliminary goal-text setting, this variant achieves performance comparable to the goal-image version. (more details in appendix G.1.)
>
> |Tool-Scene| Goal Image | Goal Text |
> |--------- |----------|----------|
> |IND|  97±3 | 95±3   |
> |OOD| 96±2    |96±3     |

---

> ### Author Response · Authors · 2025-11-25
> **Rebuttal by Authors**
>
> > Q2. Boundary artifacts in DiffCollage: Figure 2 shows DiffCollage leaving visible boundary gaps, but the original DiffCollage image-generation results generally do not show such artifacts. Could this discrepancy arise because the noisy factorization assumption breaks more severely in video or temporal domains than in static image composition? A short clarification would help reconcile this difference.
>
> Thank you for the insightful question. Yes, the discrepancy mainly comes from the fact that static image composition is a much easier setting than long-horizon video planning. DiffCollage is evaluated on static images, where factors only need to agree along short spatial boundaries；in contrast, in planning the model must maintain spatial and temporal coherence across whole trajectory. GSC sidesteps part of this difficulty by training separate models for each skill and assuming the skill sequence is given at test time (e.g., pick cube → place cube → pick hook → push cube), so the temporal structure is hand-specified rather than inferred. Our setting is strictly harder: from only start and goal, we must infer both the intermidiate skill primitives and their ordering, and enforce global consistency across them. This is exactly why we frame the problem as **message passing** on a factor graph, not merely smoothing local transitions.
>
>
> We would like to thank you again for asking important questions and suggestions about our work. We believe the additional clarifications and results have strengthened the paper. Please let us know if you have any additional concerns or questions.
>
> [1] Du, Y., Yang, M., Florence, P., Xia, F., Wahid, A., Ichter, B., Sermanet, P., Yu, T., Abbeel, P., Tenenbaum, J. B., Kaelbling, L., Zeng, A., & Tompson, J. (2023). Video Language Planning. arXiv preprint arXiv:2310.10625.
>
> [2] Xie, A., Rybkin, O., Sadigh, D., & Finn, C. (2025). Latent Diffusion Planning for Imitation Learning. arXiv preprint arXiv:2504.16925.
>
> [3] Hatch, K. B., Balakrishna, A., Mees, O., Nair, S., Park, S., Wulfe, B., Itkina, M., Eysenbach, B., Levine, S., Kollar, T., & Burchfiel, B. (2024). GHIL-Glue: Hierarchical Control with Filtered Subgoal Images. arXiv preprint arXiv:2410.20018.
>
> [4] Black, K., Nakamoto, M., Atreya, P., Walke, H., Finn, C., Kumar, A., & Levine, S. (2023). Zero-Shot Robotic Manipulation with Pretrained Image-Editing Diffusion Models. arXiv preprint arXiv:2310.10639.
>
> [5] Feng, Y., Han, J., Yang, Z., Yue, X., Levine, S., & Luo, J. (2025). Reflective Planning: Vision-Language Models for Multi-Stage Long-Horizon Robotic Manipulation. arXiv preprint arXiv:2502.16707.

---

### Meta-Review · Area_Chair_d6Q8 · 2026-01-07

**Summary:**

The paper received strong support from four of the five reviewers. These reviewers highlighted the method’s training-free, plug-and-play design, consistent improvements over strong baselines on long-horizon robotic planning tasks, and convincing validation in both simulation and real-world settings. One reviewer remained unconvinced, arguing that using Tweedie estimates for guidance is standard practice and that performance gains may stem from implementation details rather than conceptual novelty. Despite this outlier view, the overwhelming majority found the work well-motivated, technically rigorous, and empirically robust. The author has carefully addressed the low scoring comments and provided a detailed response to the reviewer's comments, answering any doubts.

**Reviewer Concerns:**

The authors provided thorough and substantive responses to all questions and concerns raised by every reviewer.

**Reviewer Scores:**

Reviewer zF3D (initial 8) would maintain their score after receiving complete technical clarifications.
Reviewer PERF (initial 6) would likely raise or maintain their score following the inclusion of CompDiffuser comparisons and DSG ablations.
Reviewer bi53 (initial 8) would maintain their score.
Reviewer dR6s (initial 2) might raise their score by 1 or 2 points given the extensive baseline validation, though they may still question the novelty of the underlying mechanism.
Reviewer geNQ (initial 8) would maintain their score.

---

### Decision · Program_Chairs · 2026-01-26

Accept (Poster)